# THIN-SHELL OBJECT MANIPULATIONS WITH DIFFERENTIABLE PHYSICS SIMULATIONS

**Yian Wang**[1]* **Juntian Zheng**[2]* **Zhehuan Chen**[3] **Zhou Xian**[4]
**Gu Zhang**[5] **Chao Liu**[6] **Chuang Gan**[1,7] †

[1]Umass Amherst    [2]Tsinghua University    [3]Peking University    [4]CMU
[5]SJTU    [6]MIT    [7]MIT-IBM

## ABSTRACT

In this work, we aim to teach robots to manipulate various thin-shell materials. Prior works studying thin-shell object manipulation mostly rely on heuristic policies or learn policies from real-world video demonstrations, and only focus on limited material types and tasks (*e.g.,* cloth unfolding). However, these approaches face significant challenges when extended to a wider variety of thin-shell materials and a diverse range of tasks. On the other hand, while virtual simulations are shown to be effective in diverse robot skill learning and evaluation, prior thin-shell simulation environments only support a subset of thin-shell materials, which also limits their supported range of tasks. To fill in this gap, we introduce *ThinShellLab* - a fully differentiable simulation platform tailored for robotic interactions with diverse thin-shell materials possessing varying material properties, enabling flexible thin-shell manipulation skill learning and evaluation. Building on top of our developed simulation engine, we design a diverse set of manipulation tasks centered around different thin-shell objects. Our experiments suggest that manipulating thin-shell objects presents several unique challenges: 1) thin-shell manipulation relies heavily on frictional forces due to the objects' co-dimensional nature, 2) the materials being manipulated are highly sensitive to minimal variations in interaction actions, and 3) the constant and frequent alteration in contact pairs makes trajectory optimization methods susceptible to local optima, and neither standard reinforcement learning algorithms nor trajectory optimization methods (either gradient-based or gradient-free) are able to solve the tasks alone. To overcome these challenges, we present an optimization scheme that couples sampling-based trajectory optimization and gradient-based optimization, boosting both learning efficiency and converged performance across various proposed tasks. In addition, the differentiable nature of our platform facilitates a smooth sim-to-real transition. By tuning simulation parameters with a minimal set of real-world data, we demonstrate successful deployment of the learned skills to real-robot settings. Video demonstration and more information can be found on the project website[1].

## 1 INTRODUCTION

Manipulating thin-shell materials is complicated due to a diverse range of sophisticated activities involved in the manipulation process. For example, to lift an object using a sheet of paper, we would instinctively create a slight bend or curve in the paper before initiating the lift (Figure 1 (a)). Human beings intuitively learn such thin-shell manipulation skills, such as folding a paper to make a crease, drawing out a piece of sheet under a bottle, and even complicated card tricks. However, existing robotic systems still struggle in handling these thin-shell objects as flexibly as humans. Compared with manipulating rigid bodies or volumetric materials, manipulating thin-shell materials poses several unique challenges. First, the physical forms of such materials are difficult to handle. For example, picking up a flat sheet is intrinsically difficult due to its close-to-zero thickness, preventing any effective grasping from the top. A more effective approach is to use friction force to

---

*Equal contribution

†Corresponding author

[1]https://vis-www.cs.umass.edu/ThinShellLab/

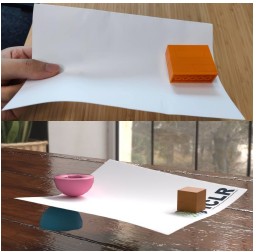 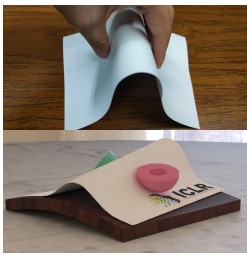 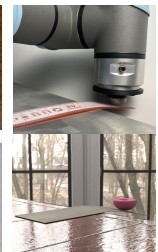 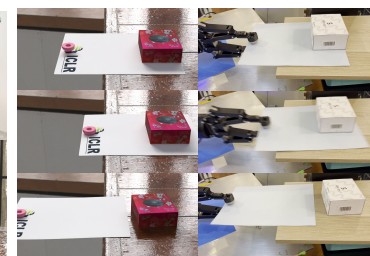

(a) Lifting a block      (b) Picking up paper      (c) sim-to-real      (d) real experiment

Figure 1: It's common for us to interact with thin-shell materials, such as bending a paper to lift a block (a) or picking up a piece of paper (b). Aiming to boost versatile robotic skill acquiring for diverse thin-shell materials, we propose *ThinShellLab*, a fully differentiable simulation platform together with a set of benchmark tasks ((a) and (b) bottom). Moreover, to bring simulation in line with the real-world, we adjust physical properties by utilizing the gradient and real-world observations (c). After that, we successfully deploy our policy learned from simulation to the real world (d).

bend a curve of the sheet before grasping it (Figure 1 (b)). Secondly, thin-shell materials are highly sensitive to even minimal variations in actions or contact points. For instance, in the lifting task (Figure 1 (a)), even millimetric movements can lead to drastically different curvatures in the paper. Furthermore, owing to the high complexity of rich contacts, gradient-based optimization methods, which have been shown to be effective and efficient in solving a wide range of deformable object manipulation tasks (Huang et al., 2021; Xian et al., 2023), suffers severely from the non-smooth optimization landscape and local optima in thin-shell manipulations.

Many prior works have investigated thin-shell manipulation tasks in real-world. Ha & Song (2021) and Xu et al. (2022b) design action primitives specifically for cloth unfolding. Zhao et al. (2023) trains robots in real-world to learn paper flipping skills. Chi et al. (2022) studies goal conditioned dynamics tasks including cloth placement and rope whipping. Namiki & Yokosawa (2015) studies paper folding with motion primitives. However, all those works are designed for solving specific tasks and are hard to scale up. Furthermore, another group of previous works build benchmarks in simulation environment to boost robotic skill learning for thin-shell materials (Lin et al., 2020; Chen et al., 2023), but they abstract away the physical contact behaviors between manipulators and objects during interacting, and ignore the bending stiffness of thin-shell materials, making them over-simplified for dexterous manipulation tasks. Some previous works also build differentiable simulators for thin-shell materials (Qiao et al., 2020; 2021; Li et al., 2021; Gan et al., 2021; Li et al., 2022), but they don't include end-effectors to manipulate thin-shells nor include benchmark tasks for thin-shell object manipulation. Finally, most of these simulation works consider thin-shell materials as cloths, which is only a subset of thin-shell materials, and neglect materials with strong bending stiffness and varying bending plasticity like papers, narrowing the scope of tasks they perform.

In this work, aiming to build a comprehensive benchmark for thin-shell manipulation skill acquiring, we propose *ThinShellLab*, a fully differentiable simulation platform for developing robotic learning algorithms with thin-shell manipulation tasks. It provides scenarios with various types of thin-shell materials with or without bending plasticity and includes the coupling among thin-shell and volumetric materials. To build such an environment, we simulate the thin-shell materials and volumetric materials following (Grinspun et al., 2003; Tamstorf & Grinspun, 2013), model the bending plasticity of creases by performing a plastic deformation on a crease when it exceeds the preset tolerance angle, implement frictional contact following (Li et al., 2020), and calculate the analytical gradient through implicit time integration scheme. To make the simulator more suitable for robot learning settings, we implement penalty-based contact to avoid time-consuming continuous collision detection and to make the gradient more smooth. To enable direct positional control of end-effectors, we apply Stable Neo-Hookean (Smith et al., 2018a) that is robust to inversion. Our engine is developed using Taichi (Hu et al., 2019), which supports massive parallel computation on GPUs or CPUs.

With the differentiable simulator, we evaluate gradient-based planning methods, sampling-based trajectory optimization methods, and state-of-the-art reinforcement learning algorithms in our manipulation task settings like block lifting (Figure 1 (a)). As also pointed out by many previous works, in our experiments, we found that reinforcement learning algorithms struggle to solve most of our tasks due to sparse rewards and high dimensions. While gradient-based methods are shown to be

effective in previous works (Li et al., 2023; Xian et al., 2023; Huang et al., 2021; Chen et al., 2023), we find it easy to stuck in local optima when working on such thin-shell manipulation due to the constant changes of contact pair, which results in different computational graph for gradient and highly non-smooth optimization landscape. To address this challenge, we propose to combine the sampling-based method and gradient-based trajectory optimization. Specifically, we use the sample-based method to search an overall policy to offer initial contact points and avoid local minima, and then apply the gradient to do refinement upon it, which would solve most of the tasks.

Besides the utilities in policy learning, we also provide inverse design tasks to find suitable material parameters to maximize certain goal functions, given fixed action trajectories. Moreover, we employ our gradients to address the sim-to-real gap by creating a simulator counterpart to the real world and fine-tuning our system parameters based on real-world observations, as depicted in Figure 1 (c). Following this process, the policy trained in our simulation can be deployed in the real world with greater ease and a reduced simulation-to-real gap (Figure 1 (d)).

We summarize our main contribution as follows:

- We introduce ThinShellLab, a comprehensive benchmark for robotic learning in thin-shell material manipulation. This benchmark is designed to facilitate the study of complex manipulation skills across a diverse array of thin-shell materials. The benchmark provides a diverse collection of manipulation tasks and a standardized interface for RL and trajectory optimization algorithms.
- At the core of ThinShellLab, we develop a fully differentiable simulation engine supporting volumetric and thin-shell deformable objects and coupling between them. To the best of our knowledge, it is the first simulation environment capable of conducting gradient-based optimization on thin-shell object manipulation tasks.
- We benchmark the performance of gradient-based, sampling-based, and hybrid trajectory optimization methods together with the state-of-the-art RL algorithms on our task collection, and show a comprehensive analysis of the challenges and behaviors of each method in different scenarios.
- Besides the manipulation tasks, we demonstrate how our differentiable simulator can help solving inverse design problem, and bridge the simulation scenes and the real-world robotics. We show the the simulator's capability of reconstructing real-world physics scene parameters via gradient, and deploying learned skills to real robots.

## 2 RELATED WORK

**Differentiable Simulation.** The graphics and robotics community has been increasingly focusing on differentiable simulation in recent years (Todorov et al., 2012; Coumans & Bai, 2016–2021; Xu et al., 2022a). Researchers have developed differentiable simulation engines for deformable objects (Chen et al., 2023; Du et al., 2021; Li et al., 2023; Huang et al., 2021) and fluids (Xian et al., 2023). There are in general two approaches of building differentiable simulation. The first approach is to train neural networks predicting forward dynamics (Schenck & Fox, 2018; Li et al., 2019; Ummenhofer et al., 2020; Han et al., 2022), which serves as a simulator and is naturally differentiable through the networks. This class of methods often requires collecting a large amount of data, and suffers from problems on generalizability and large sim-to-real gap. The second class is to explicitly build a simulation system and integrate gradient calculation into the system, through auto-diff for explicit time integration (Hu et al., 2020; Qiao et al., 2020; 2021), or by adjoint methods or implicit function differentiation for implicit time integration (Du et al., 2021; Li et al., 2022).

**Thin-Shell Object Manipulation.** Robotic manipulation on thin-shell materials is also widely investigated (Chi et al., 2022). One popular direction is garment folding or unfolding (Ha & Song, 2021; Xu et al., 2022b; Wu et al., 2023). Most of these previous works are trained in simulators in which the cloth is modeled as mass-spring systems that perform no bending stiffness (Lin et al., 2020) or learned directly from real-world data. Another line of works investigates robotic manipulation on paper, which is of greater bending stiffness compared to cloths. Zhao et al. (2023) learns how to flip bookpages using a soft gripper from real-world data. Namiki & Yokosawa (2015) studies folding a sheet of paper in the real-world by extracting motion primitives. While those previous works can indeed solve tasks in the real-world, they are designed for specific tasks and are hard to generalize. Compared with these works, our work aims to build a comprehensive benchmark for thin-shell object manipulation.

| Simulator | Integration | Diff | Bending | Bending Plasticity | Coupling | Skill Learning |
|-----------|-------------|------|---------|--------------------|----------|----------------|
| TDW | Explicit | ✓ | ✓ | | | |
| Softgym | Explicit | | | | | ✓ |
| DAXBENCH | Explicit | ✓ | | | | ✓ |
| DiffSim | Implicit | ✓ | ✓ | | ✓ | |
| DiffCloth | Implicit | ✓ | ✓ | | | |
| C-IPC | Implicit | | ✓ | | | |
| Ours | Implicit | ✓ | ✓ | ✓ | ✓ | ✓ |

Table 1: **Comparison with previous simulators.** we compare our work with other simulators supporting thin-shell materials. Here *Diff* indicates differentiability, *Bending* and *Bending Plasticity* indicates modelling of bending energy and its plasticity behavior, and *Coupling* refers to the seamless integration of thin-shell material manipulation with other dynamic objects.

For previous works that are also building simulators and benchmark tasks for thin-shell manipulation (Lin et al., 2020; Chen et al., 2023), they typically model thin-shell materials as cloths with no bending-stiffness by mass-spring model. Additionally, they abstract away the frictional contact in thin-shell material grasping, which could lead to a substantial sim-to-real gap in cases when direct grasping fails. Hence, we aim to build a simulator that can model diverse thin-shell materials with a wide range of bending-stiffness and bending plasticity, as well as frictional contact between thin-shell materials and our end-effectors to support various scenarios for robotic skill learning.

## 3 SIMULATION ENGINES

At the core of ThinShellLab, we present a fully differentiable simulation engine, implemented using Taichi programming language (Hu et al., 2019; 2020), which enables hardware acceleration on both GPUs and CPUs. The engine models a variety of thin-shell and volumetric materials, along with their interactions. In forward simulation, we use an implicit time integration scheme to simulate the dynamics of the whole system and apply Newton's method to solve the forward dynamics equation. Table 3 shows a comparison between ThinShellLab and other related simulation engines supporting thin-shell object modelling.

### 3.1 MATERIAL MODELLING

Following the traditional finite-element methods, ThinShellLab represents both volumetric and thin-shell deformable objects by spatially discretizing the objects into tetrahedral or triangular elements. At each timestep $t$, the internal interaction of deformable objects is modeled by potential energy $U(\mathbf{x}^t)$ for each type of material, where $\mathbf{x}^t$ represents the nodal positions of discretized vertices. We also calculate the derivatives $\partial U/\partial \mathbf{x}^t$ and $\partial^2 U/(\partial \mathbf{x}^t)^2$, which correspond to the internal force and force differential required by the implicit Euler solver. For volumetric materials, we maintain the deformation gradient $\mathbb{F}_e^t \in \mathbb{R}^{3\times3}$ relative to the rest configuration at each tetrahedral element $e$ and calculate the Stable Neo-Hookean (Smith et al., 2018b) energy $U_e(\mathbb{F}_e^t)$ induced by internal elastic force. For thin-shell materials, we follow the Kirchhoff-Love shell theory to define their potential energy as the sum of stretching energy $U_s(\Delta \mathbf{A}_e^t)$ and bending energy $U_b(\Delta \theta_e^t)$, which are defined on the stretching ratio $\Delta \mathbf{A}_e^t \in \mathbb{R}^4$ of each triangular element $e$'s area and edge lengths, and relative bending angle $\Delta \theta_e^t \in \mathbb{R}$ on each edge $e$ connecting two adjacent elements, related to their rest configuration $\overline{\theta}_e^t \in \mathbb{R}$. The energy and forces in thin-shell materials are modeled based on prior work (Grinspun et al., 2003; Tamstorf & Grinspun, 2013). We further model the bending plasticity by setting up a yield criterion $\kappa$ and change the rest configuration for the angle on edge $e$ if $\|\Delta \theta_e^t\| > \kappa$. More details of materials and forward simulation are shown in Appendix A.1.

### 3.2 FRICTIONAL CONTACT MODELLING

We model frictional contact behavior by potential energies defined on the nodal positions. Before every time step $t$, we detect all potential collision pairs using fast spacial hashing and express the signed distance $d_k^t \in \mathbb{R}$ and tangential relative movement $\mathbf{u}_k^t \in \mathbb{R}^2$ as functions of vertices in the $k$-th pair. A negative $d_k^t$ means penetration and therefore should be penalized. The contact potential

energy is the sum of a repulsive penalty potential $U_r(d_k^t)$ and a friction potential $U_f(\mathbf{u}_k^t)$:

$$U_r(d_k^t) = \frac{\mathbf{k}_r}{2} \max(d_k^t - \epsilon_r, 0)^2, \;\; U_f(\mathbf{u}_k^t) = \mu_k \lambda_k f_0(\epsilon_v, \|\mathbf{u}_k^t\|).$$

The penalty potential $U_r$ is a quadratic penalty function approximating the barrier $d > \epsilon_r$, in which $\epsilon_r$ models the thickness of the contact surface and $\mathbf{k}_r$ served as a stiffness parameter. The friction potential $U_f$ is a smooth friction model from Incremental Potential Contact (IPC) (Li et al., 2020), which unifies the sticking and slipping modes by a smooth approximation. Here $\mu_k$ is the friction coefficient, $\lambda_k$ is the strength of the repulsive force at the last timestep and $f_0$ is the smoothing kernel proposed by IPC, with a parameter $\epsilon_v$ of accuracy tolerance. Benefiting from the robustness of the iterative implicit solver, we can use large $\mathbf{k}_r$ and tiny $\epsilon_v$ to achieve a high accuracy, while maintaining a high numerical stability. The implementation details of the contact model are further explained in Appendix A.3.

## 3.3 GRADIENT BACK-PROPAGATION

To enable the use of gradient-based optimization methods for our benchmark tasks and parameter optimization, we've integrated a backward gradient solver into our simulation engine, making it differentiable. Our approach formulates forward simulations as optimization problems concerning energies at each time step (refer to Appendix A.1). In this framework, gradient back-propagation is treated as implicit function differentiation as outlined in DiffPD (Du et al., 2021). Within each time step, gradients of a specific objective function from the subsequent system state are propagated backward to the current step's state and material parameters. After that, we gather the nodal position gradients and convert it into gradients related to the manipulator trajectory. For more details of the calculations, please refer to Appendix A.2.

## 3.4 CONSTRAINT SIMULATION

To retain the position of a vertex during forward simulation, we enforce the corresponding dimension of the vector $\partial U / \partial \mathbf{x}^t$ and the matrix $\partial^2 U / (\partial \mathbf{x}^t)^2$ to be zero. This ensures a invariable solution for the vertex. For simulating a movable rigid body, we move it before each timestep and then fix it in place during the subsequent timesteps.

## 4 THINSHELLLAB BENCHMARK

Based on the proposed differentiable simulation engine, we present a robotic skill learning benchmark providing a diverse collection of thin-shell object manipulation tasks equipped with differentiable reward function, and a standardized interface for skill learning algorithms. We illustrate the proposed tasks in Figure 2.

### 4.1 BENCHMARKING INTERFACE

**Task formulation** In each manipulation task, the robot employs 6-DoF manipulators or 7-DoF parallel grippers. These components mimic real-world vision-based tactile sensors (see Section 5.3). We simplify the model by omitting the robot arm's link structure, as arm movement can be derived from end-effector trajectories using inverse kinematics methods. For policy learning, each manipulation task is formalized as the standard finite-horizon Markov Decision Process (MDP) with state space $\mathcal{S}$, action space $\mathcal{A}$, reward function $\mathcal{R} : \mathcal{S} \times \mathcal{A} \times \mathcal{S} \to \mathbb{R}$, and transition function $\tau : \mathcal{S} \times \mathcal{A} \to \mathcal{S}$ which is determined by forward simulation of the physics engine. We optimize the agent to devise a policy $\pi(a|s)$ that samples an action sequence maximizing the expected cumulative return $E_\pi \left[ \sum_{t=0}^{\infty} \gamma^t R(s_t, a_t) \right]$. Details of reward setting are listed in Appendix B.

**State space** As mentioned in Section 3.1, the deformation state of non-rigid objects is characterized by the positions of discretized vertices. To denote relevant quantities, we define $N_v$ as the total number of vertices, $N_e$ as the total number of bending edges, and $N_m$ as the total degrees of freedom (DoF) for all manipulators. The complete simulation state for the system is represented as $\mathbf{S} = (\mathbf{x}, \dot{\mathbf{x}}, \mathbf{r}, \mathbf{M})$, where $\mathbf{x}, \dot{\mathbf{x}} \in \mathbb{R}^{N_v \times 3}$ denote the positions and velocities of non-fixed vertices, $\mathbf{r} \in \mathbb{R}^{N_e}$ represents the rest configurations, and $\mathbf{M} \in \mathbb{R}^{N_m}$ signifies the pose configurations of manipulators.

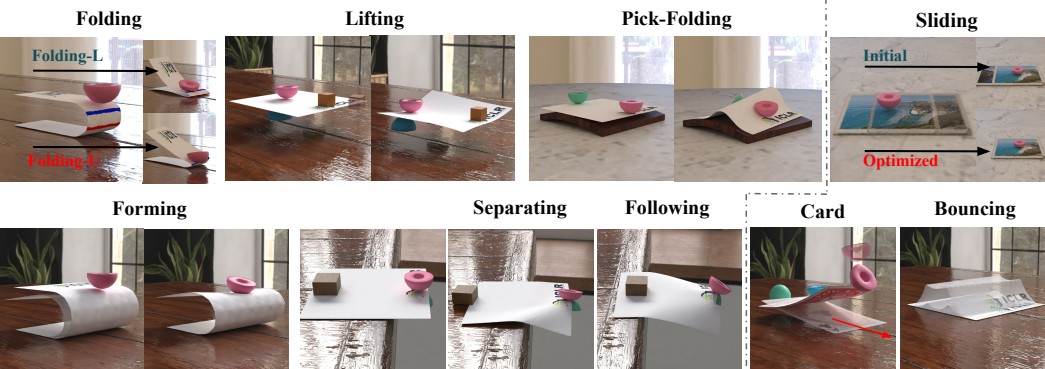

Figure 2: This figure shows 7 manipulation tasks on the left side and 3 inverse design tasks on the right side. We display the initial and the final position for our manipulation tasks and we show the target transparently in **Lifting** and **Forming**. For **Sliding**, we show the final goal with transparency and show the results before and after optimization on the right column. For **Bouncing**, we show the final state before and after optimization in transparency. We display the behavior of **Card** in transparent and draw an array representing the target moving direction of the card.

**Observation** To facilitate learning and control algorithms, we employ downsampling to reduce the dimension of observation space. We select $N_d$ points within the scene and stack their positions and velocities, forming a vector of dimensions $N_d \times 6$. We uniformly sample $N_s$ points from each sheet and $N_v$ points from each volumetric object, and therefore $N_d = N_v + N_s$. Furthermore, we incorporate the manipulators' poses into the final observation vector of dimensions $N_d \times 6 + N_m$.

**Action Space** We assume that the elastic manipulators are attached to rigid-body base parts, which have 6 DoFs for single manipulators and 7 DoFs for parallel grippers. The action space is then defined as the pose change of all manipulators.

## 4.2 MANIPULATION TASKS

**Lifting** This task begins with a flat sheet with two manipulators on the bottom, one on the top, and a block positioned on the opposite side. The objective is to raise the block using the sheet and transport it to a different location.

**Separating** Given a sheet of paper on a table, a cube resting on it, and a parallel gripper gripping the paper, the agent must pull out the paper and separate it from the cube horizontally, striving for maximum separation distance.

**Following** In a scenario identical to **Separating**, this task involves moving the cube along with the paper towards the right side as far as possible by pulling the paper.

**Folding** This task revolves around folding a sheet of paper to induce plastic deformation. The scene comprises a bending strip on the table with one end fixed and a manipulator positioned on it. The objective is to reinforce the crease, either on the upper or lower position of the curve, which split it into two tasks: **Folding-upper** and **Folding-lower** (abbreviated as **Folding-U** and **Folding-L**).

**Pick-Folding** In this task, the agent operates two manipulators initialized on opposite sides of a sheet of paper resting on an arched table. The objective is to skillfully pick up the paper and fold it, with the specific aim of creating a robust crease in the middle of the paper.

**Forming** Beginning with a curved paper on the table, one side fixed, and a gripper on top, the agent's objective is to manipulate the paper using a single end-effector to achieve a predefined goal shape. This shape is generated by executing a random trajectory and saving the final paper position.

## 4.3 INVERSE DESIGN TASKS

In this set of tasks, we maintain fixed manipulator trajectories and instead optimize system parameters to achieve specific objectives.

**Sliding** The scene consists of three sheets of paper stacked on a table, with a manipulator pressing and sliding on them. The goal is to optimize the friction coefficient between the sheets to ensure they all move together.

**Bouncing** We begin with a flat sheet on the table featuring two creases. The sheet initially bounces due to the bending force exerted by the creases. The task's objective is to optimize the bending stiffness to achieve higher bounce height.

**Card** Initialization involves a card held by three manipulators, followed by executing a card shuffling movement to launch the card. The aim in this task is to adjust the bending stiffness to project the card as far to the right as possible.

## 5 EXPERIMENT

In this section, we quantitatively evaluate methods including the state-of-the-art RL algorithms, together with sampling-based, gradient-based, and hybrid trajectory optimization on ThinShellLab's manipulation benchmark tasks, and analyze their behavior and performance on different tasks. We also experiment with inverse design and real-world tasks including real-to-sim system identification and real-world object manipulation.

### 5.1 METHOD EVALUATION WITH MANIPULATION TASKS

We conduct a comprehensive performance evaluation of various methods, including gradient-based trajectory optimization (GD), model-free Reinforcement Learning (RL) algorithms (Soft Actor-Critic - SAC (Haarnoja et al., 2018) and Proximal Policy Optimization - PPO(Schulman et al., 2017)), and a sampling-based trajectory optimization method named Covariance Matrix Adaptation Evolution Strategy (CMA-ES) (Hansen & Ostermeier, 2001). To ensure replicability, we use open-source implementations for the RL algorithms (Raffin et al., 2021) and CMA-ES (Hansen et al., 2019). For gradient-based trajectory optimization, we employ the Adam optimizer. Additionally, we explore a hybrid approach, initializing optimization with CMA-ES and then transitioning to gradient-based optimization (CMA-ES + GD). Considering the possibility that both CMA-ES and RL algorithms may require numerous iterations to converge, we report the maximum scores achieved within a specified number of episodes for performance insights. (see Table 2) We delve into task behavior to highlight the inherent challenges, and analyze the strengths and limitations of various methods in addressing these challenges.

| Tasks | Lifting | Separating | Following | Folding-U | Folding-L | Pick-Folding | Forming |
|---|---|---|---|---|---|---|---|
| PPO | $2.22 \pm 0.11$ | $6.01 \pm 0.02$ | $-0.53 \pm 0.03$ | $1.13 \pm 0.28$ | $3.32 \pm 1.19$ | $0.17 \pm 0.19$ | $2.34 \pm 0.33$ |
| SAC | $1.81 \pm 0.09$ | $7.73 \pm 0.89$ | $2.81 \pm 0.21$ | $2.43 \pm 0.02$ | $3.49 \pm 0.38$ | $0.85 \pm 0.84$ | $3.00 \pm 0.30$ |
| CMA-ES | $1.42 \pm 0.01$ | $8.08 \pm 0.18$ | $3.36 \pm 0.70$ | $3.22 \pm 0.04$ | $3.93 \pm 0.80$ | $2.14 \pm 2.14$ | $3.51 \pm 0.26$ |
| GD | **4.15** | 6.80 | -0.50 | -0.03 | 1.26 | 0.03 | 1.69 |
| CMA-ES + GD | $2.51 \pm 0.39$ | $\mathbf{8.24 \pm 0.27}$ | $\mathbf{3.43 \pm 0.89}$ | $\mathbf{3.81 \pm 0.03}$ | $\mathbf{4.68 \pm 0.17}$ | $\mathbf{4.74 \pm 2.11}$ | $\mathbf{3.63 \pm 0.14}$ |

| Tasks | Sliding | Bouncing | Card | Tasks | Sliding | Bouncing | Card |
|---|---|---|---|---|---|---|---|
| CMA-ES | $\mathbf{4.47 \pm 0.09}$ | $0.62 \pm 0.04$ | $-1.02 \pm 0.01$ | GD | $4.34 \pm 0.03$ | $\mathbf{0.75 \pm 0.01}$ | $\mathbf{-1.00 \pm 0.04}$ |

Table 2: We show the maximum reward within a fixed number of episodes and the standard deviation of it. Since GD has no randomness in different trials, it has no standard deviation.

**Task Behavior** Our investigation reveals that, despite the structural similarity of the defined **Folding** tasks, they exhibit markedly divergent behavioral patterns, with **Folding-upper** displaying notably higher complexity compared to **Folding-lower**. Specifically, in **Folding-lower**, the agent simply needs to establish contact with or below the upper curve and subsequently push in a left-downward direction. Conversely, in the case of **Folding-upper**, the agent must delicately harness both bending and friction forces to navigate the upper curve towards the right side before proceeding with the folding action. Among the tasks we examined, **Pick-Folding** poses the greatest challenge. This can be attributed to several factors: 1) the necessity for initial contact points in close proximity to the

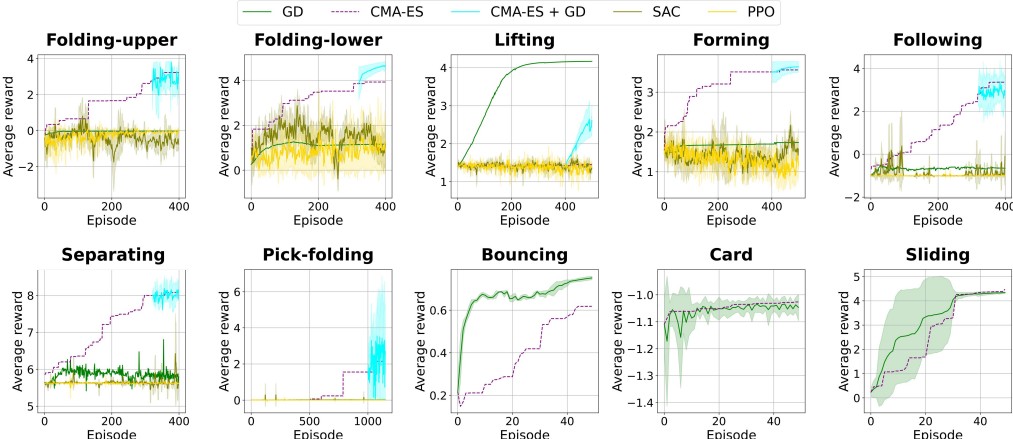

Figure 3: Reward curves for all methods. We plot the prefix maximum for the CMA-ES method. We observe that the hybrid method of CMA-ES + GD achieves the best performance in most tasks.

center; 2) the task's high reliance on precise management of friction forces; 3) the sparse reward structure associated with this task. **Separating** shows intriguing behavior: the gripper initially grabs the paper towards the block object to impart an initial speed to the block before extracting the paper. Subsequently, we demonstrated the increased efficiency of this approach in real-world scenarios. Additionally, we explored the replacement of paper with a tissue characterized by lower bending stiffness, resulting in observable variations in agent behavior. For more tasks and experiments, we include them in Appendix B.

**RL Algorithms** As depicted in Figure 3, RL algorithms, while not achieving optimal efficiency, exhibit reasonable performance in tasks characterized by straightforward behaviors, such as **Folding-lower** and **Forming**. Given their nature of random searching, these methods generally yield higher maximum scores compared to direct GD, which often encounters local optima in most tasks. However, it is important to acknowledge the inherent limitations of RL algorithms. 1) RL algorithms are notably deficient in sample efficiency, requiring a considerable number of steps to converge. 2) They demonstrate limited progress in tasks that demand fine-grained control, as evident in **Lifting**, or when tasks become progressively more detailed, such as **Forming**. 3) RL algorithms face significant challenges when confronted with demanding tasks, such as **Pick-Folding**, which features sparse rewards, or **Folding-upper**, characterized by intricate behavioral requirements.

**CMA-ES** CMA-ES exhibits reasonable performance across most tasks, effectively evading local minima and providing improved initial policies for gradient-based methods. However, it encounters challenges **Lifting**. Although it manages to acquire the ability to bend the paper to prevent the block from falling, it struggles to advance the block closer to its intended destination. This difficulty can be attributed to the intricate interplay of the three end-effectors, coupled with the sensitivity of the task's dynamics, rendering the sample-based CMA-ES method less suited for this particular challenge. Furthermore, our observations indicate that CMA-ES experiences a decelerated learning curve in later episodes and requires an extended number of episodes to achieve convergence. This characteristic, while beneficial for searching for an initial policy, proves less suitable for addressing finer-grained tasks that demand more rapid progress and adaptation.

**Gradient-based Trajectory Optimization** In all the tasks considered, our gradients effectively guide the policy towards improved performance within the initial time steps. However, it is noteworthy that this process often becomes ensnared in local optima. This phenomenon is especially evident in the **Folding** tasks, as illustrated in Figure 3. We attribute this behavior to the limitations of the gradient-based method, which does not account for changes in contact pairs and is sensitive to even minimal alterations in the action sequence.

**Hybrid method** To address local optima and enhance fine-grained control, we combined the sample-based and gradient-based methods. Initially, the sample-based approach identifies contact points and establishes an initial policy. Subsequently, the gradient-based method fine-tunes the trajectory. This hybrid approach excels in most manipulation tasks, except for **Lifting**, where it eventually converges to a similar reward level as the gradient-based method, albeit less efficiently.

## 5.2 RESULTS OF INVERSE DESIGN TASKS

For inverse tasks, we evaluate CMA-ES and GD methods. As evident in Figure 3, both the gradient-based and sample-based methods perform effectively in these tasks. However, the gradient-based method exhibits a faster and smoother growth in inverse design tasks, benefiting from its guidance by physical principles. Notably, in the **Card** and **Sliding** tasks, the standard deviation of the GD method diminishes over time. This trend indicates that, despite commencing with distinct initial system parameters, our gradient-based method converges these parameters towards a uniform outcome.

While the **Bouncing** task necessitates a higher bending stiffness and the **Sliding** task requires an increased friction coefficient, the **Card** task poses a more intricate challenge. In the case of the **Card** task, the bending stiffness must strike a delicate balance. If it is too high, the card reaches a greater distance but tends to flip out prematurely due to limited curvature by the end-effectors, resulting in a shorter overall moving distance. Furthermore, altering the bending stiffness can lead to changes in contact pairs, contributing to a larger standard deviation in the later time steps of the **Card** task compared to the other two inverse design tasks.

## 5.3 REAL-WORLD EXPERIMENT

In this section, we leverage the differentiability of our simulator to fine-tune physics parameters according to real-world data. We further conduct experiments of transferring skills learned from the simulator to real-world robotic systems, showing the capability of ThinShellLab's learning framework to bridge between simulation and real-world scenarios.

**Real-to-sim System Identification.** In the real-to-sim process, we initiate real-world manipulation experiments employing a robotic system equipped with force and tactile sensors. Subsequently, we construct corresponding simulation environments that integrate a differentiable loss function, quantifying the disparities between simulation outcomes and real-world results. Leveraging the differentiability of our physics engine, we back-propagate gradients to adjust material parameters and employ gradient descent methods for optimization, as outlined in Section 5.2. Our real-to-sim system identification tasks encompass **Tactile**, **Friction**, and **Bending**, serving as demonstrations of ThinShellLab's capacity to reconstruct real-world scenes from collected data. For comprehensive results and additional details, we refer readers to Appendix C.

**Real-world Manipulation Experiment.** After reconstructing physics parameters, we transferred the **Separating** manipulation task from simulation to a real-world robotic system. We optimize the task in the virtual scene with reconstructed parameters and deploy it to a real scenario with a robotic arm and parallel gripper. Our optimized trajectory, as described in Section 5.1, successfully generated an initial velocity on the object to prevent sticking during sheet retraction, proving effective in the real world. In contrast, a straightforward heuristic policy of dragging at a constant speed failed due to insufficient relative speed. For more details, please refer to Appendix D.

## 6 CONCLUSION AND FUTURE WORK

We introduced ThinShellLab, a comprehensive robotic learning benchmark for thin-shell object manipulation based on a fully differentiable simulation engine, presented with a standardized interface for skill learning methods and a set of benchmark tasks. Through these benchmark tasks, we identify challenges like non-continuous optimization landscapes due to complex contacts and high material sensitivity for thin-shell manipulation. To tackle these challenges, we found that a hybrid approach, combining sample-based and gradient-based optimization, performs best across most tasks. We conducted real-world experiments, fine-tuning system parameters to match real-world observations, and successfully deployed our learned skills in real-world scenarios.

While we are working on the real-to-sim process to address the simulation-to-real gap, our current approach relies only on tactile and force sensors. However, it's important to note that there might still be a residual gap due to our contact model and inherent real-world instability, leading to random errors. Hence, future research could explore the integration of visual observations and advanced simulation-to-real techniques to further minimize this gap.

## ACKNOWLEDGEMENT

We thank Tiantian Liu for the insightful discussions and help with the experiments. Special appreciation goes to Meshy, where the first author completed an internship and conducted a portion of this research. We thank the anonymous reviewers for their helpful suggestions. This work is funded in part by grants from Google, Amazon, Cisco, Toyota Motor North America, and Mitsubishi Electric Research Laboratories.

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

## A  IMPLEMENTATION DETAILS

### A.1  FORWARD SIMULATION

We apply implicit time-stepping scheme in our simulation engine. Let $n$ be the total DoF of the whole system and $\mathbf{x}^t, \mathbf{v}^t \in \mathbb{R}^n$ be the generalized coordinates and velocities at time-step $t$. Then the implicit time integration would be formulated as:

$$\mathbf{x}^{t+1} = \arg\min_{\mathbf{x}} \mathbf{g}(\mathbf{x}) = \arg\min_{\mathbf{x}} \frac{1}{2h^2}\|\mathbf{x} - \mathbf{y}^t\|_{\mathbf{M}}^2 + U(\mathbf{x}),$$

in which $\mathbf{y}^t = \mathbf{x}^t + h\mathbf{v}^t + h^2\mathbf{M}^{-1}\mathbf{f}_{ext}$ and $\mathbf{M}$ is the mass matrix of the system, $\mathbf{f}_{ext}$ is the total external force. This is equivalent to the following gradient form (Du et al., 2021):

$$\mathbf{0} = \nabla\mathbf{g}(\mathbf{x}) = \frac{1}{2h^2}\mathbf{M}(\mathbf{x} - \mathbf{y}^t) - \mathbf{f}_{int}(\mathbf{x}),$$

in which $\mathbf{f}_{int}$ refers to the internal forces, i.e., $-\nabla U(\mathbf{x})$. Since $\nabla\mathbf{g}(\mathbf{x})$ is in general non-linear, this equation is solved iteratively using Newton's method at each time step. To fix the $j$-th variable in simulation, we only need to set the $j$-th value of vector $\nabla\mathbf{g}(\mathbf{x})$ to zero and set the $j$-th row and column of its Hessian matrix to zero in Newton's method. Further details of energy calculation will be discussed in Appendix A.4.

## A.2 GRADIENT BACK-PROPAGATION

We can back-propagate the gradient through the implicit function $\nabla \mathbf{g}(\mathbf{x}) = \mathbf{0}$. As also mentioned in DiffPD (Du et al., 2021), we calculate the derivative between $\mathbf{x}^t$, and $\mathbf{x}^{t+1}$ and $\mathbf{x}^{t-1}$ to deliver the gradient of the loss function $\frac{\partial L}{\partial \mathbf{x}^{t-1}} = \frac{\partial L}{\partial \mathbf{x}^{t+1}} \frac{\partial \mathbf{x}^{t+1}}{\partial \mathbf{x}^{t-1}} + \frac{\partial L}{\partial \mathbf{x}^t} \frac{\partial \mathbf{x}^t}{\partial \mathbf{x}^{t-1}}$. Note that we also have to calculate the loss gradient for $\partial \overline{\theta}^t$ in all the time steps, since we are modeling the plasticity of thin-shell materials. To start with, we first rewrite the function as follows:

$$\mathbf{0} = \nabla \mathbf{g}(\mathbf{x}) = \frac{1}{2h^2} \mathbf{M}(\mathbf{x} - 2\mathbf{x}^t + \mathbf{x}^{t-1} - h^2 \mathbf{M}^{-1} \mathbf{f}_{ext}) - \mathbf{f}_{int}(\mathbf{x}, \mathbf{x}^t_{fixed}, \mathbf{x}^t, \overline{\theta}^t, \eta),$$

where $\mathbf{x}^t$ and $\mathbf{x}^{t-1}$ are node positions of $t_{th}$ and $(t-1)_{th}$ timestep, $\overline{\theta}^t$ are undeformed angles for $t_{th}$ timestep, $\mathbf{x}^t_{fixed}$ are positions of fixed nodes in the beginning of $(t+1)_{th}$ timestep, and $\eta$ is the material parameters. Here, $\overline{\theta}^t$ is actually a function of $\mathbf{x}^t$ and $\overline{\theta}^{t-1}$, and the friction force is a function of $\mathbf{x}_i$ and $\mathbf{x}_{i-1}$. We then back-propagate the loss gradient through this implicit function to obtain gradients for each time step. These gradients, applied to a subset of $\mathbf{x}^t_{fixed}$ corresponding to the manipulator's rigid base part, are aggregated to the gradient of manipulator trajectories, as mentioned in Section 4.1.

## A.3 COLLISION DETECTION AND CONTACT HANDLING

The collision detection is done by fast grid hashing of the contact surfaces. We spatially split the whole 3D space into grids. For each triangular face in a object, we store its information in grid that the triangle's circumcentre is occupying. When querying the contact state of an incoming vertex, we search in the nearby $3 \times 3 \times 3$ grids and detect collisions with all triangles involved in these grids. In order to ensure the correctness of hashing, we set the edge size of grids to be greater than half of the maximal diameter of all triangles.

Since our contact handling uses quadratic penalty potential instead of strict barrier functions, we design a mechanism to avoid crashes caused by penetration between vertices and thin-shell objects. Once a collision has been detected, we record the normal direction of contact surface, and then keep it unchanged until the collision state is removed. In this way, the contacting vertices will not penetrate to the opposite side of the thin-shell object, and therefore make the simulation robust and stable.

While the logarithmic barrier employed in IPC imposes strict non-penetration, it tends to significantly slow down convergence in Newton's method. Moreover, it demands continuous collision detection during the solution of forward dynamics and can introduce instability into gradients during back-propagation. Hence, given that extreme accuracy isn't always imperative in robotics tasks, we simplify the calculation of the penalty function by retaining the normal direction of contact pairs. This approach remains stable even in scenarios where penetration occurs.

## A.4 ENERGY CALCULATION

### A.4.1 CONTACT ENERGY

According to Section 3.2, the contact energies $U_r$ and $U_f$ are defined as

$$U_r(d) = \frac{k_r}{2} \max(d - \epsilon_r, 0)^2, U_f(\mathbf{u}) = \mu \lambda f_0(\epsilon_v, \|\mathbf{u}\|),$$

where the distance $d$ is a signed distance between vertex $\mathbf{p}$ and its projection to a triangle $(\mathbf{p}_1, \mathbf{p}_2, \mathbf{p}_3)$, whose normal direction is represented by $(\mathbf{p}_2 - \mathbf{p}_1) \times (\mathbf{p}_3 - \mathbf{p}_1)$. $k_r$ stands for a constant stiffness parameter, $\mu$ stands for a friction coefficient, and $\lambda$ stands for the contact force

along normal direction at last timestep. The signed distance $d$ is then calculated as

$$d = \frac{(\mathbf{p} - \mathbf{p}_1) \cdot [(\mathbf{p}_2 - \mathbf{p}_1) \times (\mathbf{p}_3 - \mathbf{p}_1)]}{\|(\mathbf{p}_2 - \mathbf{p}_1) \times (\mathbf{p}_3 - \mathbf{p}_1)\|}$$

$$= \frac{(\mathbf{p} - \mathbf{p}_1) \cdot \det[\mathbf{p}_2 - \mathbf{p}_1, \mathbf{p}_3 - \mathbf{p}_1, (\hat{\mathbf{x}}, \hat{\mathbf{y}}, \hat{\mathbf{z}})]}{\|(\mathbf{p}_2 - \mathbf{p}_1) \times (\mathbf{p}_3 - \mathbf{p}_1)\|}$$

$$= \frac{\det[\mathbf{p}_2 - \mathbf{p}_1, \mathbf{p}_3 - \mathbf{p}_1, \mathbf{p} - \mathbf{p}_1]}{\|(\mathbf{p}_2 - \mathbf{p}_1) \times (\mathbf{p}_3 - \mathbf{p}_1)\|},$$

in which the 3D vertices are represented as column vectors. Let $\mathbf{x} = [\mathbf{p}_1, \mathbf{p}_2, \mathbf{p}_3, \mathbf{p}] \in \mathbb{R}^{12}$ denote the involved variables, we implemented tool functions in Taichi to calculate

$$D(\mathbf{x}) = \det[\mathbf{p}_2 - \mathbf{p}_1, \mathbf{p}_3 - \mathbf{p}_1, \mathbf{p} - \mathbf{p}_1],$$
$$C(\mathbf{x}) = \|(\mathbf{p}_2 - \mathbf{p}_1) \times (\mathbf{p}_3 - \mathbf{p}_1)\|,$$

together with the gradients $\partial D/\partial \mathbf{x}, \partial C/\partial \mathbf{x}$ and Heissan matrices $\partial^2 D/\partial \mathbf{x}^2, \partial^2 C/\partial \mathbf{x}^2$. With the help of SymPy library for symbolic computation, we optimized the computation graph and hard coded it into Taichi snippets. The repulsive energy $U_r$ is then a composition function on $D(\mathbf{x}), C(\mathbf{x})$, and therefore we use the chain rule to calculate its gradient and Hessian matrix.

For the friction energy $U_f(\mathbf{u})$, we follow IPC (Li et al., 2020) to calculate the $f_0(\epsilon_v, \|\mathbf{u}\|)$ function, its derivative $f_1(\epsilon_v, \|\mathbf{u}\|)$ and second derivative $f_2(\epsilon_v, \|\mathbf{u}\|)$. Once we get the $\partial U_f/\partial \|\mathbf{u}\|$ and $\partial^2 U_f/\partial \|\mathbf{u}\|^2$, we transform it back to the tangential space to get $\partial U_f/\partial \mathbf{u}$ and $\partial^2 U_f/\partial \mathbf{u}^2$.

### A.4.2 ELASTIC ENERGY

We follow Stable Neo-Hookean (Smith et al., 2018b) to define the elastic energy as

$$U_e(\mathbf{F}) = \frac{\mu}{2}(I_C - 3) - \mu(J - 1) + \frac{\lambda}{2}(J - 1)^2,$$

in which $\mathbf{F}$ denotes the deformation gradient of one tetrahedral element, $I_C = \text{tr}(\mathbf{F}^T\mathbf{F})$, and $J = \det \mathbf{F}$. The motivation for replacing the original $\log J$ term in the Neo-Hookean model with $J - 1$ is that $J - 1$ is stable even for $J \leq 0$. Therefore, even if the whole tetrahedral element is inverted, this Stable Neo-Hookean model still works robustly and can provide gradients and Hessian matrices to recover the element (see the original paper of Stable Neo-Hookean (Smith et al., 2018b)). This situation is common in our simulator, since the robot manipulator involves a moving boundary and can easily cause inversion when the moving speed is large. We observe that the Stable Neo-Hookean perfectly addresses this problem.

The gradient $\partial U_e/\partial \mathbf{F}$ can be computed easily. To compute the Hessian matrix $\partial^2 U_e/\partial \mathbf{F}^2$, we first enumerate the differential $\delta F_{ij}$ for single entry and then calculate the correponding force differential $\delta(\partial U_e/\partial \mathbf{F})$. We follow a well-known tutorial on FEM (Sifakis & Barbic, 2012) to calculate the force differentials, and then concatenate them into the Hessian matrix $\partial^2 U_e/\partial \mathbf{F}^2$.

### A.4.3 THIN-SHELL ENERGY

Following Grinspun et al. (2003), we apply the stretching energy $U_s$ and the bending energy $U_b$ for thin-shell materials. We further divide the stretching energy into $U_s^e$ and $U_s^a$, representing stretching energy for edges and areas of triangles. The energy formula is as follows:

$$U_s^e = \sum_e K_e(1 - |e|/|\bar{e}|)^2 |\bar{e}|,$$

$$U_s^a = \sum_A K_a(1 - |A|/|\overline{A}|)^2 |\overline{A}|,$$

$$U_b = \sum_e K_b(\theta_e - \overline{\theta_e})^2 |\bar{e}|/|\overline{h_e}|,$$

where $e$ represents all the edges, $A$ represents all the triangles, and $\theta_e$ represents the angle on edge $e$. $|\bar{e}|$, $|\overline{A}|$, $|\overline{h_e}|$, and $\overline{\theta_e}$ respectively represent the rest length of the edge, the rest area for the triangle,

one-third of the average height of the two adjacent triangles of the edge, and the rest angle on this edge. $K_e$, $K_a$, and $K_b$ are the coefficients.

For $U_s^e$, it's simple to calculate the derivatives and Hessian matrix. The derivative of the length $|e|$ to the nodal point positions $x_0$ and $x_1$ is $\frac{\partial|e|}{\partial x_0} = (x_0 - x_1)/|e|$. The Hessian matrix will require the calculation of $\frac{\partial^2|e|}{\partial a_0^2}$ and $\frac{\partial^2|e|}{\partial a_0 \partial b_0}$ due to symmetry, where $x_0 = (a_0, b_0, c_0)$.

For $U_s^a$, it becomes more complex to compute the derivatives and Hessian matrix. Assume the triangle is $(x_0, x_1, x_2)$ and $x_0 = (a_0, b_0, c_0)$. Similar to the calculation of the contact energy, we use math tools to generate the symbolic formula of the derivative $\frac{\partial|A|}{\partial a_0}$ and hard code it with Taichi; then, we can calculate all the derivatives due to symmetry. For the Hessian matrix, we use math tools to generate formulas for $\frac{\partial^2|A|}{\partial a_0^2}$, $\frac{\partial^2|A|}{\partial a_0 \partial a_1}$, $\frac{\partial^2|A|}{\partial a_0 \partial b_0}$, $\frac{\partial^2|A|}{\partial a_0 \partial b_1}$; then, we can calculate the full matrix using symmetry.

For $U_b$, the Hessian matrix is too complex to compute, even when we try to use some math tools. As a result, we follow Tamstorf & Grinspun (2013) to compute the derivatives and Hessian matrix for bending energy. This paper discusses how to compute the Hessian matrix of bending energy in a simplified form.

### A.4.4  BACK-PROPAGATION

This part has been briefly discussed in Appendix A.2. The equation to be solved in forward simulation is

$$\mathbf{0} = \nabla \mathbf{g}(\mathbf{x}) = \frac{1}{2h^2}\mathbf{M}(\mathbf{x} - 2\mathbf{x}^t + \mathbf{x}^{t-1} - h^2\mathbf{M}^{-1}\mathbf{f}_{ext}) - \mathbf{f}_{int}(\mathbf{x}, \mathbf{x}_{fixed}^t, \mathbf{x}^t, \overline{\theta}^t, \eta), (1)$$

where $\mathbf{x}^t$ and $\mathbf{x}^{t-1}$ are node positions of $t_{th}$ and $(t-1)_{th}$ timestep, $\overline{\theta}^t$ are undeformed angles for $t_{th}$ timestep, $\mathbf{x}_{fixed}^t$ are positions of fixed nodes in the beginning of $(t+1)_{th}$ timestep, and $\eta$ is the material parameters. Here, $\overline{\theta}^t$ is actually a function of $\mathbf{x}^t$ and $\overline{\theta}^{t-1}$, and the friction force is a function of $\mathbf{x}_i$ and $\mathbf{x}_{i-1}$. DiffPD (Du et al., 2021) discusses the general idea of how to pass gradient in the implicit time integration:

$$\frac{\partial \nabla \mathbf{g}(\mathbf{x})}{\partial \mathbf{x}}\frac{\partial \mathbf{x}}{\partial \mathbf{y}} + \frac{\partial \nabla \mathbf{g}(\mathbf{x})}{\partial \mathbf{y}} = \frac{\partial}{\partial \mathbf{y}}\mathbf{0}$$

$$\frac{\partial}{\partial \mathbf{x}}[\frac{1}{h^2}\mathbf{M}(\mathbf{x} - \mathbf{y}) + \nabla \mathbf{E}(\mathbf{x})]\frac{\partial \mathbf{x}}{\partial \mathbf{y}} + \frac{\partial}{\partial \mathbf{y}}[\frac{1}{h^2}\mathbf{M}(\mathbf{x} - \mathbf{y})] = 0$$

$$[\frac{1}{h^2}\mathbf{M} + \nabla^2\mathbf{E}(\mathbf{x})]\frac{\partial \mathbf{x}}{\partial \mathbf{y}} - \frac{1}{h^2}\mathbf{M} = 0$$

$$\frac{\partial \mathbf{x}}{\partial \mathbf{y}} = \frac{1}{h^2}[\nabla^2\mathbf{g}(x)]^{-1}\mathbf{M}$$

With $\frac{\partial \mathbf{x}}{\partial \mathbf{y}}$ in hand, we can backpropagate the gradient from $x^{t+1}$ to $y^t$. Given $y^t = 2x^t - x^{t-1} + h^2M^{-1}f_{ext}$, we can deliver the gradient to both $x^t$ and $x^{t-1}$.

Similarly, by substituting $y$ in the above equation with other variables like $\mathbf{x}_{fixed}^t$ or $\overline{\theta}^t$, we can calculate the gradient. The remaining challenge is computing the term $\frac{\partial \nabla \mathbf{g}(\mathbf{x})}{\partial \mathbf{y}}$ in the first line when changing $y$ into other variables. For these variables, this term will involve the Jacobian matrix for the force, as shown in Equation (1).

For $\overline{\theta}^t$ and $\eta$, their relationship to the forces is linear, allowing us to calculate the Jacobians directly. Regarding $\mathbf{x}_{fixed}^t$, the Jacobians are computed during forward simulation, where they are set to zeros. We can apply the same procedure in forward simulation and retain the Jacobians during the backpropagation process.

Concerning $\mathbf{x}^t$, it primarily influences the friction force since we utilize the pressure force, contact surface, and the original projected point from the last timestep. We observe that Huang et al. (2022) calculates the derivative for friction similarly to our approach. The distinction lies in our choice not to pass the gradient through the surface change during backpropagation.

| Task | $\mu_{object}^{table}$ | $\mu_{cloth}^{table}$ | $\mu_{cloth}^{manipulator}$ | $\mu_{cloth}^{object}$ | bending stiffness | action range |
|---|---|---|---|---|---|---|
| Lifting | - | - | 5.0 | 5.0 | 100 | 0.001 |
| Separating | 0.0 | 0.2 | 5.0 | 0.2 | 100 | 0.002 |
| Following | 0.0 | 0.2 | 5.0 | 0.2 | 100 | 0.002 |
| Folding-U | - | 5.0 | 5.0 | - | 400 | 0.001 |
| Folding-L | - | 5.0 | 5.0 | - | 400 | 0.001 |
| Pick-Folding | - | 0.1 | 5.0 | - | 200 | 0.001 |
| Forming | - | 5.0 | 5.0 | - | 200 | 0.001 |
| Sliding | - | 0.4 | 1.0 | - | 1000 | - |
| Bouncing | - | 0.5 | - | - | - | - |
| Card | - | 1.0 | 1.0 | - | - | - |

Table 3: Detailed parameters.

## A.5 RENDERING

We have customized LuisaRender (Zheng et al., 2022), a ray-tracing rendering framework, to facilitate the rendering of our simulation results. This tailored version of the renderer boasts support for a wide range of material models and possesses the capability to generate highly realistic rendering outcomes.

# B DETAILS OF MANIPULATION TASKS

## B.1 TASK DETAILS

In RL algorithms, we employ downsampling of nodal points to construct the observation input.

In all our tasks, we consistently utilize a simulation step of 5e-3 seconds, and we establish a maximum action range specific to each task to ensure system stability. For the majority of our tasks, we confine the action range to 1 millimeter per timestep. However, for tasks that inherently demand higher speeds, such as the **Separate** and **Following** tasks, we extend the action range to 2 millimeters.

In the case of RL and CMA-ES, we employ a heuristic approach to truncate the action range. In contrast, for gradient-based trajectory optimization, we employ a loss function to softly regulate action speed. These adjustments help maintain the stability and effectiveness of our reinforcement learning algorithms in various tasks.

We further list our detailed parameter for each task in Table B.1.

## B.2 REWARD

The positive direction of the x-axis is defined as pointing towards the interior of the table for all tasks.

**Lifting** The reward for successful completion is determined by taking the negative sum of the squared distances between each point of the block and its target position. For better distinguish the difference we use $-lg(-reward)$ as the new reward when plotting and filling the table.

**Separating** The reward for this task is computed as the difference between the sum of the x-coordinates of the block and the sum of the x-coordinates of the paper, multiplied by the ratio of the total number of nodal points on the block to that on the paper.

**Following** In the context of this task, the reward is calculated as the negative average x-coordinate of the block.

**Folding** For **Folding-upper**, the reward is determined as the sum of the upper crease rest angles minus the sum of lower crease rest angles, whereas for **Folding-lower**, it is the opposite.

**Pick-Folding** The reward for this task is the sum of crease angles for the edges located in the middle of the sheet.

**Forming** The reward for this task is the negative sum of the squared distances between each point on the paper and its respective target position. Similar to **Lifting** we apply $-lg(-reward)$ as the new reward.

**Sliding** The reward for this task is determined as the negative sum of the x-coordinates of the bottom paper.

**Bouncing** The reward function for this task is defined as the sum of the z-coordinates of the points located on two creases.

**Card** In this task, the reward is defined as the negative sum of the x-coordinates for every nodal point on the card.

The loss is generally negative rewards in those tasks, while for GD method we include another loss to limit the action range.

### B.3 MORE ANALYSIS

Although we attempted to guide the learning process in **Pick-Folding** by introducing a small reward for increasing the angle of the middle curve and observing slight improvements in gradient-based methods, the optimization process frequently became trapped in local optima. Furthermore, sample-based trajectory optimization also grappled with finding a viable solution, with some trials failing to identify a suitable initial solution, leading to significant variance in the final reward across different trial runs. Nonetheless, it is noteworthy that the CMA-ES + GD method demonstrated rapid learning capability when provided with a reasonably well-initialized solution for **Pick-Folding**. This may be attributed to the task's dynamic simplification after the sheet is picked up.

We list the detailed design and analysis of the hybrid method as follows.

- **Parameters:** We opt for a population size of 40 in CMA-ES with zero-initialized trajectories. The initial variance is set to 1.0, corresponding to 0.0003 in each dimension of manipulator movement per timestep. The maximum movement within one timestep is either 0.001 or 0.002, depending on the task. Generally, we execute CMA-ES for 80 percents of the total episodes, where episodes are defined by the number of rollouts. For the Pick-Folding task, which demands more extensive training, we set the CMA-ES episode count to 1000 and supplement it with an additional 150 episodes of gradient descent.

- **Design:** To enhance the effectiveness of the CMA-ES method, we filter out non-feasible states, such as instances where there is no contact between manipulators and thin shell or when the manipulator is experiencing excessive force. In these situations, we directly assign a low reward value and save the time for executing the rest actions.

- **Analysis:** Some curves in Figure 3 exhibit large variance. Specifically, in **Folding-U**, **Following**, and **Separating**, the variance arises from the sensitivity of thin-shell manipulation results to minor changes, as previously discussed in the introduction section. In the case of **Pick-Folding**, although sensitivity contributes to the variance, the primary factor is occasional failure of the CMA-ES method. Failures result in rewards near 0, and successful runs yield rewards greater than 6, leading to substantial variance in the outcomes.

### B.4 SUCCESS INDICATOR

Defining the level of success in certain tasks proves challenging due to uncertainty regarding the optimal performance achievable within the current settings using the best policy. Currently, we gauge the success of a policy by visually inspecting the results. Nevertheless, we aim to enhance analytical insight by presenting the scores of human-designed heuristic policies alongside theoretically optimal scores. Subsequently, we can formulate a success indicator based on an analysis of these scores.

- **Human Designed** To elucidate the meaning of reward values, we present the scores of human-designed policies for each task in the table below. Notably, generated policies occasionally out-

perform human-designed counterparts, particularly in tasks requiring meticulous manipulation and where certain policies exhibit unexpected effectiveness (as observed in **Separating**).

- **Oracle** In addition to human-designed policies, we also present the theoretically maximum scores for each task, disregarding certain physical constraints. For **Pick-Folding**, we calculate this score by assuming the entire middle curve is fully folded. In the case of **Separating**, we assume the paper is affixed to the manipulator, while the block experiences no friction force with the paper. For **Following**, we assume both the manipulator and the block are affixed to the paper. In both **Folding-L** and **Folding-U**, we assume that the target curve is fully folded while the other one is completely flattened. For tasks involving **Lifting** and **Forming**, we assume a perfect match with the target. It is important to note that although certain Oracle scores, such as those for **Pick-Folding** and **Separating**, are exceptionally high, they are **not physically feasible within the current experimental settings**. We present

- **Success Score** To make it further concrete, we design the score of success for each task. Concretely, we define the score of success as $\min(\text{HumanDesign}, \text{Oracle}/2)$ for **Pick-Folding**, **Separating**, and **Following**, and $\text{Oracle}/2$ for **Folding-U** and **Folding-L** since human-designed policies perform poorly in those two tasks. The successful score of **Forming** and **Lifting** is $-3 \times 10^{-4}$.

- **Success Rate** Finally, we calculate the success rate of our CMA-ES + GD method according to the mean and standard deviation.

|  | Pick-Folding | Separating | Following | Folding-L | Folding-U | Forming | Lifting |
|---|---|---|---|---|---|---|---|
| Human Design | 5.49 | 7.23 | 5.05 | 1.63 | -0.24 | 0.0 | -0.001 |
| Oracle | 42.24 | 20.47 | 7.35 | 6.42 | 6.42 | 0.0 | 0.0 |
| Success | 5.49 | 7.23 | 3.68 | 3.21 | 3.21 | -0.0003 | -0.0003 |
| Success rate | 0.36 | 1.0 | 0.39 | 1.0 | 1.0 | 0.76 | 1.0 |

Table 4: Success Indicator.

## C    REAL-TO-SIM SYSTEM IDENTIFICATION TASKS

We conduct three real-to-sim system identification tasks **Tactile**, **Friction** and **Bending** to assess ThinShellLab's capability of reconstructing real-world scenes. In the **Tactile** task, we push a vision-based tactile sensor to a flat desktop and then optimize the Young's modulus and Poisson's ratio of the elastomer using collected data of exerted force and shifting of surface dot matrix. In the **Friction** task, we push the same tactile sensor to a sheet of paper and then drag the paper to create slipping friction between it and the desktop. We optimize the friction coefficient to make the simulated friction force matched with collected data. Finally, the **Bending** task involves a piece of thin-shell sheet object with non-negligible bending stiffness, such as poker card, card paper or metallic ruler, whose stiffness parameter is to be identified. We fix the object on the desktop's edge, and then push it using a tactile sensor as shown in Figure 1(c). We optimize the bending stiffness to match the simulated and collected force data. Figure 4 shows the optimization curve of these tasks. Figure 5 shows the dot matrix of tactile sensor after optimizing the **Tactile** task.

We provide additional insights into the **Tactile** task, as illustrated in Figure 6. Initially, we perform an action sequence involving **Pressing Down** and subsequently optimize the Young's modulus and Poisson's ratio of the elastomer using collected data on exerted force and the shifting of the surface dot matrix. Following this optimization, we further refine the system by optimizing the friction coefficient through the execution of **Pressing and Sliding**. Once these parameters are determined, we evaluate the outcomes in a novel action sequence involving **Pressing and rotating along the x-axis**. The results indicate minimal discrepancies in this new action sequence.

In the **Bending** task, we have encountered challenges in accurately retrieving small-scale bending stiffness from the real world using tactile sensors. This difficulty arises from the sensitivity of results to the Poisson ratio of the tactile material.

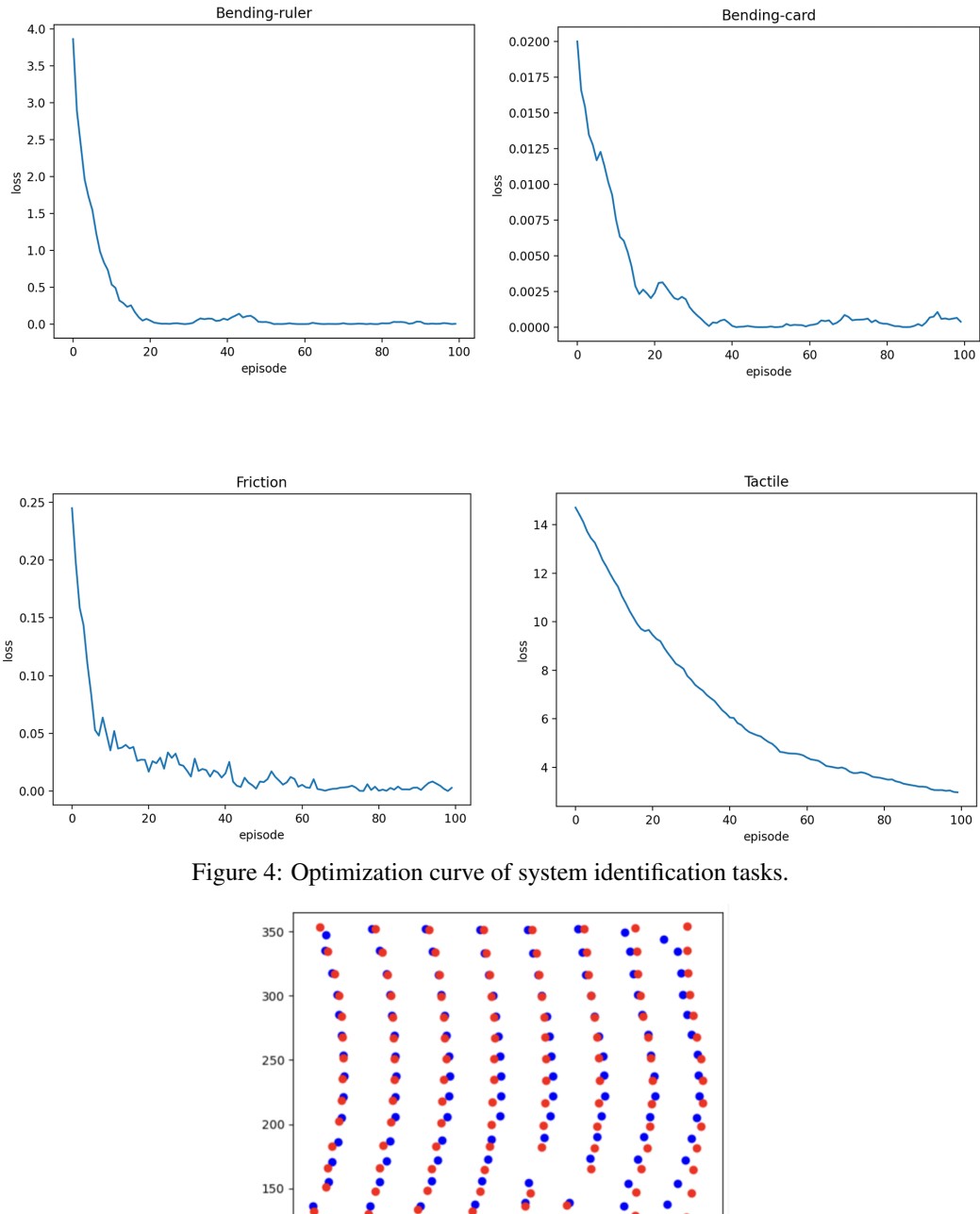

Figure 4: Optimization curve of system identification tasks.

Figure 5: Tactile dot matrix after optimization.

## D REAL-WORLD MANIPULATION EXPERIMENT

We transfer the **Separating** and **Folding** tasks into the real world. In the **Folding** tasks designed to align with the real-world, we optimize it with a rigid manipulator with 3 DoFs (which we deem is enough to solve the task). As illustrated in Figure 7, we successfully execute both the **Separating** and **Folding** tasks in a real-world setting. Upon comparing the simulation results with the real-world executions, we observe highly similar behaviors, albeit not precisely identical, indicating a minimal sim-to-real gap. Notably, in the **Separating** task, the model generates an initial velocity, showcasing the successful approximation of bending stiffness in our real-to-sim process, where the

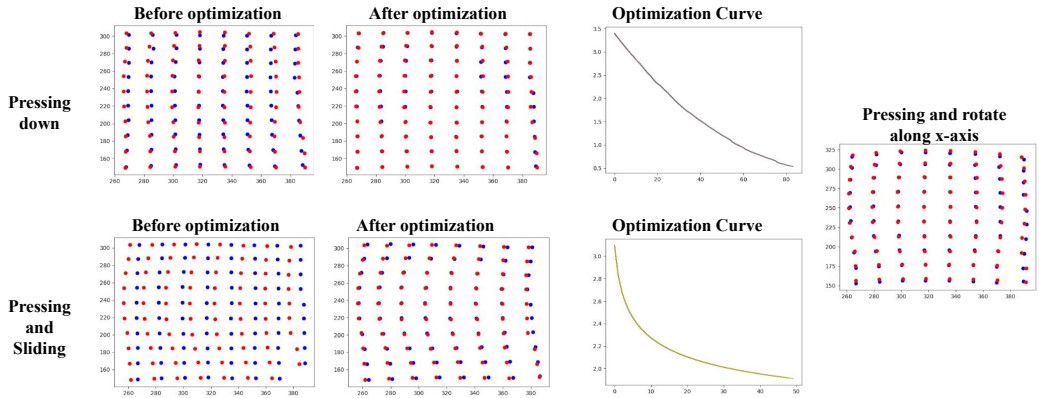

Figure 6: Real-to-sim tactile task. We show the final stage of the tactile marker before and after optimization. The red points are the 2d marker position in real world and the blue points are in simulator.

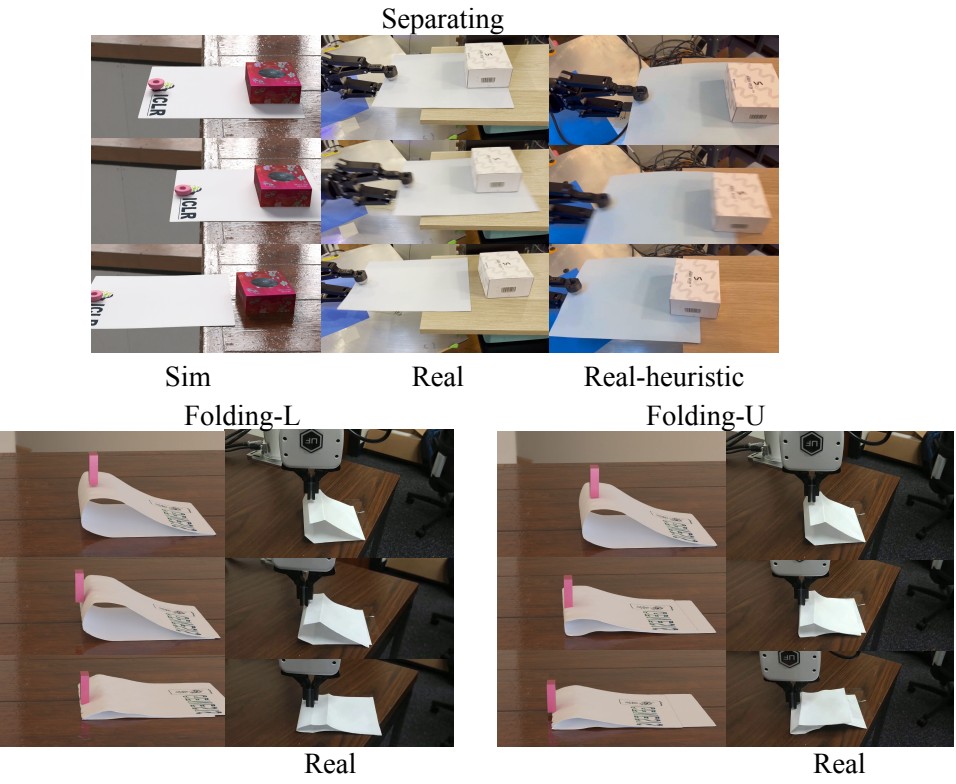

Figure 7: Real-world manipulation results.

bending stiffness is estimated to be sufficiently large to impart velocity to the object. The success in the **Folding** task further demonstrates the model's awareness of the dynamic properties of the paper, a result attributed to the effectiveness of our real-to-sim process.

## E SPEED PERFORMANCE

In ThinShellLab, both the forward and backward passes are parallelized on a single GPU. The major bottlenecks of the running time are **1) linear equation solving at each iteration of Newton's method** and **2) Hessian matrices building and SPD projecting**. For Hessian matrices, we use the Taichi programming language to process the submatrices for independent elements in parallel. We also implement the QR iteration in Taichi to project each submatrix onto a manifold of SPD

(symmetric positive definite) matrices. For linear equation solving, we use a Python binding of the cuSPARSE library, which supports solving sparse matrices using Cholesky Factorization on GPU. The remaining parts of the simulation (e.g., line search and collision detection) are fast enough and therefore can be negligible. Collision detection is efficiently solved by space hashing using Taichi's sparse data structure.

To benchmark the running speed of the simulation, we test our simulator on the *Forming* task, which contains a typical scene with 1698 DoFs involving both volumetric and thin-shell deformable bodies. The test is done on a single RTX 4090. On average, the forward/backward passes take **36.41 s / 4.67 s** respectively for 100 timesteps, i.e., framerates of **2.74 fps / 21.41 fps**. In comparison, DiffCloth reports framerates at **0.73 fps / 2.72 fps** using the same Newton-Cholesky method and **7.14 fps / 17.54 fps** using the fast PD acceleration, both under a scene with 2469 DoFs. C-IPC reports a framerate at **0.024 fps** for forward simulation of a thin-shell scene with 18.3K DoFs.

In practice, it takes about 3 hours to train a policy for one task.

While we run our experiments on a single GPU, it is also possible to run multiple environments on different GPUs in parallel. We didn't do this because gradient-based trajectory optimization methods do not benefit from parallel environments.

## F    FUTURE DIRECTIONS

Through the evaluation of various methods in our benchmark tasks of thin-shell manipulation, we have concluded that reinforcement learning algorithms exhibit limited data efficiency. This inefficiency arises from the sensitivity of thin-shell materials and the heightened complexity of their states. Simultaneously, gradient-based methods face challenges in reliability within contact-rich scenarios.While sample-based methods, such as CMA-ES, generally perform well, they suffer from reduced data efficiency in later episodes and intricate settings. Consequently, we propose a hybrid approach that combines the strengths of CMA-ES and gradient-based methods to achieve superior results.

In terms of future work, we recommend exploring closed-loop control under varied parameters, initializations, and tasks. Currently, our focus has primarily been on open-loop methods like CMA-ES and gradient-based methods. We posit that an avenue worth exploring involves leveraging the synergy between reinforcement learning and gradient methods (Mora et al., 2021) for improved data efficiency in closed-loop control. Additionally, an alternative strategy could involve collecting successful trajectories under diverse settings using open-loop methods and subsequently employing a neural network for imitation learning. This approach aims to address generalization challenges rather than optimization problems.

