# OpenReview forum: "Thin-Shell Object Manipulations With Differentiable Physics Simulations"
_ICLR.cc/2024/Conference — ICLR 2024 spotlight_

### Official Review · Reviewer_k5Lf · 2023-10-28

**Soundness:** 4 excellent
**Presentation:** 3 good
**Contribution:** 4 excellent
**Rating:** 8
**Confidence:** 4

**Summary:**

This work develops ThinShellLab, a simulator and benchmark for robotic learning in thin-shell material manipulation.
The simulator models volumetric and thin-shell materials with finite-element methods with tetrahedral and triangular elements, and is differentiable.
The robotic manipulation benchmark defines tasks including lifting, separating, following, folding, pick-folding, and forming. The benchmark is used to evaluate manipulation methods including sampling-based, gradient-based, and hybrid trajectory optimization, as well as reinforcement learning methods.
The simulator is also shown to support system identification, which allows transferring certainly simulated policies to be deployed in the real world.

**Strengths:**

- This work is the first to properly implement a differentiable thin-shell object simulator for robotic manipulation.
- The paper is overall written clearly.
- The manipulation tasks are well-designed to reflect properties of thin-shell materials. The benchmarked methods have good coverage.

**Weaknesses:**

- Some details of the simulator are not clear (see Questions section).
- Some useful simulation features are not shown in this paper.
  - It seems the system does not fully support/couple with rigid-body simulation, which is important for modeling robots.
  - The paper does not show any tasks involving constraint simulation (e.g., cloth with one end fixed).
- Real-world manipulation only contains a single case. It is not convincing that in general the described system identification method will improve sim-to-real policy deployment.

**Questions:**

- The contact model uses quadratic energy instead of barrier. How does the system correctly identify that 2 triangles are penetrating? Based on the paper, it seems it is based on normal direction, however, this does not seem correct since the normal would flip if one of the contacting shell is flipped.
- The paper describes the system as simulating tetrahedral and triangular elements, how is the rigid-body action space achieved?
- Newton's method requires solving a large linear system. How is this implement on the GPU? Is it solved through sparse matrix algorithms, conjugate gradient methods, or some other method?
- What is the speed/performance of this simulator? What are the bottlenecks?

---

> ### Author Response · Authors · 2023-11-18
> **Response to Reviewer k5Lf**
>
> *Thank you for your insightful and constructive comments! We discuss some of your questions and concerns below.*
>
> **1. Rigid body dynamics & action space**
>
> While we can directly integrate the rigid body dynamics into the implicit-time integration scheme, the system will become quite cumbersome and therefore we decide to process the deformable and rigid system separately. Currently the robot manipulator is modelled by position-based control, i.e., the action space of rigid body end-effectors are defined as SE(3) movements within each timestep, and therefore the elastomers are attached to moving rigid boundaries controlled by the actions.
>
> We are still working on further modelling of rigid body dynamics. For the rigid manipulated objects, we plan to support them by adding an stiff orthogonality potential energy on a deformable mesh as in ABD[1]. We also plan to include the penalty energies of joint constraints to model the dynamics behavior of articulated body system.
>
> **2. Constrainted simulation**
>
> We support the constraints of fixing vertex positions. In **Folding** and **Forming**, we fix one end of the paper on the table. In **Separating** and **Following**, we fix two manipulator in a gripper shape.
>
> The technique to simulate the constraints is simple. To fix one vertex (or even one dimension of this vertex), we simply set the derivative and hessian matrix related to this dimension to zero before solving the equation. This will result in an unchanged position for this vertex while modeling it's force and influence to other vertices.
>
> **3. Real-world manipulation**
>
> We add more real-world manipulation cases identical to **Folding** tasks in simulation. We use a rigid manipulator to interact with a curved paper and try to fold on different creases.
>
> ### Questions
>
> > **Q1: The contact model uses quadratic energy instead of barrier. How does the system correctly identify that 2 triangles are penetrating? Based on the paper, it seems it is based on normal direction, however, this does not seem correct since the normal would flip if one of the contacting shell is flipped.**
>
> To model contact behavior, we apply quadratic penelty energies on a dynamic set of contact constraints. The reason of not using a barrier function is that we observe that the quadratic model converges much faster than the log-barrier model on complex scenes involving multiple layers of thin-shell material. With quadratic energies, we actually make non-penetration a soft constraint with a large stiffness coefficient. When a constraint is established between a vertex and a shell surface, we maintain a "side" property between them, whose value is one of the shell's two sides. This property remains unchanged until the vertex's projection falls on the shell's boundary. When dealing with contact between the vertex and the shell, all faces of the shell will be assigned with a normal direction pointing towards the corresponding side. In this way, we avoid the ambiguity problem of shell surface's normal direction, and also the penetration problem caused by the soft constraint, since a vertex slightly penetrating into a surface will be repulsed towards a consistent direction.
>
> > **Q2: The paper describes the system as simulating tetrahedral and triangular elements, how is the rigid-body action space achieved?**
>
> It is addressed above in point 1.
>
> > **Q3: Newton's method requires solving a large linear system. How is this implement on the GPU? Is it solved through sparse matrix algorithms, conjugate gradient methods, or some other method? What is the speed/performance of this simulator? What are the bottlenecks?**
>
> For linear equation solving, we use a python binding of the cuSPARSE library, which supports solving sparse matrices using Cholesky Factorization on GPU. The speed is around 2.74 FPS in one case and the bottlenecks are **1) linear equation solving at each iteration of Newton's method** and **2) Hessian matrices building and SPD projecting**.
> The details could be found in general response point 4.
>
>
> [1] Lan, Lei, et al. "Affine body dynamics: Fast, stable & intersection-free simulation of stiff materials." arXiv preprint arXiv:2201.10022 (2022).
>
> *We sincerely appreciate your comments. Please feel free to let us know if you have further questions.*
>
> Best,
> Authors

---

> > ### Comment · Reviewer_k5Lf · 2023-11-21
> >
> > Thank you for the clarification and it has answered my questions resolved my concerns. I will keep the accept recommendation.

---

### Official Review · Reviewer_BtTD · 2023-10-30

**Soundness:** 4 excellent
**Presentation:** 3 good
**Contribution:** 4 excellent
**Rating:** 8
**Confidence:** 4

**Summary:**

This study focuses on teaching robots to manipulate a variety of thin-shell materials. It includes the introduction of ThinShellLab, a fully differentiable simulation platform that is specifically designed for robotic interactions with an array of thin-shell materials. The researchers conduct numerous experiments using reinforcement learning (RL) algorithms, in conjunction with sampling-based, gradient-based, and hybrid trajectory optimization methods, across 10 different thin-shell manipulation tasks.

**Strengths:**

1. The ThinShellLab is unique, standing as the first simulator of its kind to support a diverse range of thin-shell materials while also maintaining differentiability.

2. Unlike existing simulators, it has the capability to support Bending Plasticity.

3. This innovative simulator offers a benchmark for understanding prevalent methods associated with thin-shell objects, accommodating various types of approaches such as RL, and trajectory optimization.

**Weaknesses:**

1. The overall presentation is good. However, there could be improvements in clearly labeling the three system identification inverse tasks - Sliding, Bouncing, and Card - in both Table 2 and Figure 3. Their current placement together without clear labels within the table or graph is confusing.

2. Furthermore, it would enhance the paper if some clear conclusions were drawn, along with providing some directions or suggestions to guide readers to explore each type of method.

**Questions:**

Interested to know what is the speed of the simulation? Is multiple GPU parallelization supported by the simulation?

---

> ### Author Response · Authors · 2023-11-18
> **Response to Reviewer BtTD**
>
> *We appreciate your positive and insightful comments! Below, we address your concerns in detail.*
>
> **1. Labeling inverse tasks**
>
> We have separate those tasks in tables and figures in our main paper. Thank you for pointing them out!
>
> **2. Conclusion and future directions**
>
> We draw a conclusion that RL methods are of low data efficiency in most of the times and it would be good to seek a combination of sample-based methods (maybe RL) and gradient based method. More discussions about this are shown in General Response point 7.
>
> ### Questions
>
> > **Q1: Interested to know what is the speed of the simulation? Is multiple GPU parallelization supported by the simulation?**
>
> The speed is approximately 2.74 FPS in a test case. Currently, there are no obstacles preventing us from running parallel environments on different GPUs. However, we haven't pursued this option because the GD method does not derive any benefits from parallelization. More details can be found in General Response point 4.
>
> *We sincerely appreciate your comments. Please feel free to let us know if you have further questions.*
>
> Best,
> Authors

---

### Official Review · Reviewer_j6Cr · 2023-10-30

**Soundness:** 4 excellent
**Presentation:** 3 good
**Contribution:** 4 excellent
**Rating:** 8
**Confidence:** 3

**Summary:**

The paper shows a comprehensive benchmark framework for thin-shell object manipulation. A simulation environment is provided that expands upon previous work by including bending-stiffness and frictional contact. Since this simulation is differentiable, the authors show how gradient-based optimization can be used to enhance existing gradient-free methods for finding better performing policies for various robotics tasks. Real-world validation is shown on a specific subset of tasks for sim-to-real accuracy.

**Strengths:**

A much needed framework for robotic manipulation of general thin-shell objects is proposed by the authors, which will enable the robotics community to tackle more challenging problems in the future. Although the separate parts in the simulation is not a novelty in itself, the whole framework and its validation in benchmarks is a valuable addition to the field. The paper describes the tasks clearly, and the accompanying website is easy to understand and shows the core contributions well. A lot of analysis was provided as well to all the tasks, and on how and why the hybrid optimization method helps in solving the given problems.

**Weaknesses:**

The presentation in the paper does not feel quite as refined as the website, especially the figures, feel like they could use some work. For example, Figure 2 does not have the image bounding boxes align, and within the Sliding subfigure you can see how the edges of the images don't match up. A similar small issue is present in Figure 3 with a typo in "Separatering". Unfortunately, most of the presented tasks are dynamic, hence images don't capture them all that well, and hence the animations on the website do a more convincing job on why this particular application case matters.
Another point is that a lot of information is pushed to the Appendix, even though it could be beneficial to have in the main text. For example, the description of the hybrid method, which is a quite large engineering contribution of the paper to show working policies, is described on a high-level. A more detailed description would help here, and mentioning how much work it was to tune/balance the gradient-free and gradient-based steps could help the community. Another example is how the real-to-sim system identification is performed, how the disparity between simulation and reality is computed, these details should be included.

**Questions:**

How well does the simulation environment run in practice, is it achieving something close to real-time, or how long does simulation take? When parallelizing the simulations on GPU, does that only benefit CMA-ES and RL, or is GD also gaining some advantage from the parallelized environments?

---

> ### Author Response · Authors · 2023-11-18
> **Response to Reviewer j6Cr**
>
> *We appreciate the positive and insightful comments from you! We adress your concerns in details below.*
>
> **1. Typo and figure quality**
>
> Thank you for pointing them out, we have made the revisions.
>
> **2. Real-to-sim disparity**
>
> While we are using tactile sensor and force feedbacks as observations in the real-world, we compute the disparity by the L2 loss between the simulated tactile markers (projected into 2D images given the camera parameters) and the images collected in real-world, and by the L2 loss between the simulated forces and the real-world force feedbacks. The loss of several experiments and 2D images of markers are ploted in the appendix.
>
> **3. Hybrid method details**
>
> This information has been incorporated into General Responses point 5. In essence, we apply a fixed proportion to the steps of CMA-ES and GD in the hybrid method. Additionally, we introduce specific constraints to shape the behavior and enhance the learning process of CMA-ES.
>
> ### Questions
>
> > **Q1: How well does the simulation environment run in practice, is it achieving something close to real-time, or how long does simulation take?**
>
> We test the speed under a specific setting, and it's 2.74 FPS for forward simulation. Details are included in General Response point 4.
>
> > **Q2: When parallelizing the simulations on GPU, does that only benefit CMA-ES and RL, or is GD also gaining some advantage from the parallelized environments?**
>
> The feature of massive parallelism provided by Taichi is designed to parallelize the computing process within a single environment, resulting in faster speeds. Consequently, all algorithms benefit from this improvement.
>
> It's worth noting that it's also possible to run multiple environments on different GPUs, which provides advantages for CMA-ES and RL (while GD does not gain advantages from this feature). In our testing, we ensure a fair comparison by using a single GPU to run a single experiment.
>
> *We additional explanations have convinced you of the merits of our work. Please do not hesitate to contact us if you have other concerns.*
>
> *We appreciate your time! Thank you so much!*
>
> Best,
> Authors

---

> ### Comment · Reviewer_j6Cr · 2023-11-21
>
> Thank you for the extensive replies to all the reviewers, and for grouping the concerns into specific categories. Excellent clarification of the solver that was implemented. The detailed runtimes now make the usability very clear for practical users of the framework. Thank you to the authors for the extra time on the rebuttal!

---

### Official Review · Reviewer_3BVm · 2023-11-02

**Soundness:** 3 good
**Presentation:** 2 fair
**Contribution:** 3 good
**Rating:** 8
**Confidence:** 3

**Summary:**

The paper introduces ThinShellLab, a differentiable simulation platform tailored for robotic interactions with diverse thin-shell materials possessing varying material properties, enabling flexible thin-shell manipulation skill learning and evaluation. The experiments highlight unique challenges in thin-shell manipulation, such as reliance on frictional forces, sensitivity to minimal variations in interactions, and frequent contact pair changes. The authors also study a hybrid optimization scheme that combines sampling-based trajectory optimization and gradient-based optimization. They also showcase the proposed simulation platform allows for a smooth transition from simulation to real-world robot deployment.

**Strengths:**

1. The proposed simulator can model diverse thin-shell materials with a wide range of bending-stiffness and bending plasticity, as well as frictional contact between thin-shell materials and end-effectors, which is hardly modeled in prior benchmarks.

**Weaknesses:**

1. The writing is not clear enough. Please see the *Questions* section for details.
2. Figure 2 can be improved. The images are not well aligned. There is not a clear boundary between "left" and "right" mentioned in the caption, which makes it hard to distinguish 7 manipulation tasks and 3 inverse design tasks.
3. It is better to have some success indicators for tasks in the benchmark. Otherwise, it is hard for readers to understand difference in performance between methods, if only rewards are provided (e.g., Table 2).

Minor typo: In B.1, the table is linked to the section B.1 instead of Table 3.

**Questions:**

1. In Sec 4.1, it seems that the observation includes a set of points. Does the order of points matter? How do the learning algorithms (SAC, PPO) used in the paper handle the observation if the input observations (points) are unordered?
2. In Sec 4.1, does the 6DoF stand for the pose change in SE(3), and does the 7DoF include one more DoF for gripper closeness (mimic behaviors for two finger tips)?
3. In Sec 5.1, the authors mentioned that they reported "the maximum scores achieved within a specified number of timesteps". Does it mean that the benchmark currently focuses on "optimization" rather than "generalization"? Besides, how do the authors count "timesteps" for CMA-ES and GD methods?
4. What's the meaning of "episode" (x-axis) for GD and CMA-ES in Figure 3?
5. In Sec 5.2, for "our gradient-based method converges these parameters towards a uniform outcome.", do the authors mean that the method converges to similar solutions despite different initialization?

---

> ### Author Response · Authors · 2023-11-18
> **Response to Reviewer 3BVm**
>
> *We appreciate the positive and constructive comments from you! We have modified our paper according to your comments.*
>
> **Success Indicators**
> While we design continuous rewards without a distinct boundary between success and failure and primarily assess the success of a policy through visualization, we also present scores for human-designed policies, theoretical analyses, and success indicators in General Responses point 6.
>
> ### Questions
>
> > **Q1: In Sec 4.1, it seems that the observation includes a set of points. Does the order of points matter? How do the learning algorithms (SAC, PPO) used in the paper handle the observation if the input observations (points) are unordered?**
>
> The order of points matters in the observation. As we uniformly sample points in the full state, the sequence of selected nodal points remains constant within a given task.
>
> > **Q2: In Sec 4.1, does the 6DoF stand for the pose change in SE(3), and does the 7DoF include one more DoF for gripper closeness (mimic behaviors for two finger tips)?**
>
> Yes, the 6DoF stand for the pose change in SE(3) (x, y, z, roll, pitch, yaw), and the 7DoF include one more DoF for gripper closeness.
>
> > **Q3: In Sec 5.1, the authors mentioned that they reported "the maximum scores achieved within a specified number of timesteps". How do the authors count "timesteps" for CMA-ES and GD methods?
> What's the meaning of "episode" (x-axis) for GD and CMA-ES in Figure 3?**
>
> **The "episode" means a rollout which is a full sequence of actions from the initial state to a final state.**
>
> The "timesteps" here has the same meaning of the previous "episode". As a result, we can simply count the number of full action sequences executed in GD, CMA-ES, and RL to count the episode. We will change them into a uniform one to avoid confusing. Thank you for pointing it out.
>
>
> > **Q4: Does it mean that the benchmark currently focuses on "optimization" rather than "generalization"?**
>
> The experiment setting we currently test on indeed focuses on "optimization" rather than "generalization".
>
> However, we still want to share vision about how to address generalization in General Response point 7. Concretely, we can switch the initial state and parameters in one task and evaluate close-loop policies. Another way is to use open-loop method to collect successful trajectories under different initial state, parameters or even different tasks, then we can use imitation learning with those data to obtain a generalist policy.
>
> > **Q5: In Sec 5.2, for "our gradient-based method converges these parameters towards a uniform outcome.", do the authors mean that the method converges to similar solutions despite different initialization?**
>
> Yes. While the **Card** still have a respectively large standard deviation in the end, it is clearly shown in **Sliding** that they converges to similar solutions.
>
>
> *Please let us know if you have any further questions for our paper. We really appreciate your time! Thank you!*
>
> Best,
> Authors

---

> > ### Comment · Reviewer_3BVm · 2023-11-20
> >
> > Thank the authors for the detailed response. It resolves my concern, and I will increase my rating.

---

### Official Review · Reviewer_Vd5A · 2023-11-05

**Soundness:** 3 good
**Presentation:** 3 good
**Contribution:** 3 good
**Rating:** 8
**Confidence:** 3

**Summary:**

The paper introduces a differentiable simulator designed to handle thin shell materials such as sheets of paper and cloth to explore manipulation of such objects. To this end a set of tasks are defined to challenge various methods. Finally, an approach to tackle the proposed tasks using combining sampling and optimization is proposed.

**Strengths:**

- Development of a simulator for specific kind of object geometries
- Definition of a set of tasks to challenge manipulation methods on thin objects
- Overall easy to understand and follow

**Weaknesses:**

The idea of the paper are clear and the writing is easy to follow. Some of that, however, stems from a lack of detail regarding the simulator and the tasks. The description of the simulator, one of the core contributions, is barely a page long and mostly covers high-level concepts, physical quantities being modeled, and methods used to implement the simulator. I would have expected to see greater detail about this as currently it's hard to assess if this is a straight forward task of writing up a few equations or involved required complex derivations and developments. What also would be good is a discussion regarding the chosen way of modeling thin objects, as the approach selected is likely not the only one and a discussion of the pros and cons of modeling the interactions one way or another would be a valuable contribution.

Another aspect where more detail would be expected are the task descriptions. The tasks are easy to understand at a high level. However, their actual definition is omitted which makes it unclear what objective the various methods are required to optimize for later on. As there are many ways to formulate the described tasks it is important to have this information present.

One aspect of the experiments that could be improved is conveying what the goal is. The only information is that the goal is to evaluate the performance and behavior of the simulator and methods. Being more concrete and actually laying out the things to be investigated would improve the readability of the entire section.

The presentation of the results is at times incomplete. For example, Table 2 shows numbers with +/- values but there is no mention of what those values are. One can assume that it is mean +/- standard deviation, however, this should be stated clearly. In that table there is also a single row which has no +/- values, why? There are also various cells in the table that are lacking values, which is not explained. The results list score values for each of the tasks, but without knowing level of "success" these scores correspond to it is impossible to know whether the differences in scores are significant or not.

The presentation of the real-world experiments does not convey any information and fails to provide information. This section should either be improved to add actual experimental data, as opposed to referring the reader to an appendix with a handful of plots, and provide a discussion.

The paper describes a hybrid approach, though details are extremely limited. Due to the lack of detail of that approach and the focus being about the simulator, it is unclear why this was added nor that it adds anything. The idea of combining global search with local refinement is sensible. However, I would expect this to be common knowledge in the RL community given the (by default) unguided nature of RL exploration and the fact that RL tends to perform well once it finds a solution to a problem. There are also odd behaviors of the hybrid method that are not discussed. In some of the curves shown in Figure 3 the variance in the scores are excessively large, why does this occur?

Another question that should be addressed is the benchmark aspect. The paper states that the goal is for this to be used as a benchmark. Is the idea to use this similar to gym and atari game setups, or more like the benchmarks used in the computer vision community with tasks on withheld data to counteract overfitting on a set of tasks?

The core aspects of the paper, i.e. the simulator and task descriptions, are good yet could benefit from additional detail. The experiments are ok, though need to be improved to more clearly convey their information.

**Questions:**

- What solution quality / success rate do the various rewards correspond to?
- How fast does the simulator run, is it real-time, faster, slower?
- What is the limit of object complexity that can be simulated?

---

> ### Author Response · Authors · 2023-11-18
> **Response to Reviewer Vd5A**
>
> *Thank you for your insightful and constructive comments! We have added additional experiments and modified our paper according to your comments.*
>
> **1.Simulation details**
>
> This is addressed in General Response point 3. The detailed equations are too extensive to include in the main paper.
>
> **2.What is the level of success for each task**
>
> As demonstrated in the General Response point 6, we integrate the theoretically highest scores with human-intuitive policies to derive the success score for each task. The success rate is then computed based on the mean and standard deviation for the CMA-ES + GD method.
>
> **3.Actual definition of task objective**
> Although we have detailed the reward function in Appendix B.2, we provide the exact formulas of the reward functions for further clarification.
>
> **Lifting**: Let $B$ be the set of vertex positions on the block and $B^T$ be the target vertex positions, and $B[i]$ be the $i_{th}$ vertex in $B$.
> $reward=-\sum_{i}\{(B[i].x - B^T[i].x)^2 + (B[i].y - B^T[i].y)^2 + (B[i].z - B^T[i].z)^2\}$
>
> **Separating**: Let $B$ be the set of vertex positions on the block and $C$ be the set of vertex positions on the paper. Since there are $256$ vertices in the shell and $144$ vertices in the block, we multiply the sume of block vertices with $256 / 144$.
> $reward=256 / 144 * \sum_{i}B[i].x - \sum_{i}C[i].x$
>
> **Following**: $reward=-\sum_{i}B[i].x$
>
> **Folding-U**: Let U be the set of rest angle (angle at rest position) on the upper crease and L be the set of rest angle on the lower crease.
> $reward=\sum_{i}U[i] - \sum_{i}L[i]$
>
> **Folding-L**: $reward=-\sum_{i}U[i] + \sum_{i}L[i]$
>
> **Pick-Folding**: Let M be the set of rest angle (angle at rest position) on the middle of the paper.
> $reward=\sum_{i}M[i]$
>
> **Forming**: Let $S$ be the set of vertex positions on the shell and $S^T$ is the target vertex positions, and $S[i]$ be the $i_{th}$ vertex in $S$.
> $reward=-\sum_{i}\{(S[i].x - S^T[i].x)^2 + (S[i].y - S^T[i].y)^2 + (S[i].z - S^T[i].z)^2\}$
>
> **Sliding**: Let $S$ be the set of vertex positions on the bottom shell.
> $reward=-\sum_{i}S[i].x$
>
> **Bouncing**: Let $S$ be the set of vertex positions on the shell.
> $reward=\sum_{i}S[i].z$
>
> **Card**: Let $C$ be the set of vertex positions on the card.
> $reward=-\sum_{i}C[i].x$
>
> **4.The presentation of the results**
>
> The symbol $\pm$ in the table denotes the standard deviation resulting from multiple runs. Notably, the GD method exhibits no standard deviation as it lacks randomness in its multiple executions. In the context of inverse design tasks, the optimization is solely focused on the parameters. Consequently, we evaluate a distinct set of methods for these tasks, which explains the absence of certain rows in the table.
>
> We will ensure to articulate these details more explicitly in the main paper. Thank you for bringing this to our attention.
>
> **5.Conveying the goal**
>
> Our goal of the experiment section is to delve into task behavior to highlight the inherent challenges, and analyze the strengths and limitations of various methods in addressing these challenges. This analysis serves as  the groundwork for the future development of innovative algorithms for thin-shell manipulation tasks.
> We've add this to our main paper, thank you for metioning it.
>
> **6.Real world experiment**
>
> * **[Real-to-sim system identification]** We have introduced more detailed experimental results for the real-to-sim system identification tasks [Appendix C]. These results are analyzed through the observation of the 2D marker pose differences, presented qualitatively, and the loss curve, presented quantitatively. Although tactile simulation is not the primary focus of this paper, we argue that it demonstrates satisfactory alignment with the real world.
> We have encountered challenges in accurately retrieving small-scale bending stiffness from the real world using tactile sensors. This difficulty arises from the sensitivity of results to the Poisson ratio of the tactile material. We deem that a better way to estimate small-scale bending stiffness is to first use vision method to obtain it's state and subsequently estimating it using the balance between gravitational and bending forces.
>
> * **[Sim-to-real manipulation tasks]** In the real-world execution of the **Separating** task, our paper discusses that the bending stiffness of the thin shell significantly influences the final policy. Successfully deploying this policy to the real world serves as evidence of our effective approximation of the bending stiffness of the card paper.
> We have also incorporated additional real-world manipulation results. In the **Folding** task, we employ a rigid manipulator with 3 Degrees of Freedom (DoFs) and train a new policy in the simulator, which is subsequently deployed into the real world. The outcome illustrates our success in capturing the dynamics of the paper and accomplishing the task in the real-world setting.

---

> ### Author Response · Authors · 2023-11-18
> **part 2**
>
> **7.Hybrid method**
> In this work, we focus more on how to utilize the gradient in simulation to train an open-loop policy within limited time. While RL policy can generalize better to unseen state, open-loop methods employing differentiable physics prove to be more data-efficient. CMA-ES, GD, and our hybrid method all operate as open-loop trajectory optimization approaches. The concept of combining CMA-ES and GD with differentiable physics is proposed to alleviate the issue of local minima associated with the GD method in scenarios characterized by rich contact interactions, while design of RL policy considers no physics properties.
> More details of this hybrid method is shown in the general response.
>
> ### Questions
>
> > **Q1: The paper states that the goal is for this to be used as a benchmark. Is the idea to use this similar to gym and atari game setups, or more like the benchmarks used in the computer vision community with tasks on withheld data to counteract overfitting on a set of tasks?**
>
> It is similar to gym and atari game setups, where users can generate their own data and build their own experiment.
>
> > **Q2: What solution quality / success rate do the various rewards correspond to?**
>
> As outlined in General Response point 6, we have established a set of scores to signify success in our benchmark. It is crucial to note, however, that we consider these tasks to be continuous rather than discrete. Our primary method of assessing success revolves around visualizations, and the success indicators are primarily employed for comparative purposes.
>
> > **Q3: How fast does the simulator run, is it real-time, faster, slower?**
>
> It's about 2.74 FPS as discussed in the General Response point 4.
>
> > **Q4: What is the limit of object complexity that can be simulated?**
>
> We employ 3D meshes to simulate objects, enabling the simulation of any object that can be represented using a 3D mesh. It's important to note, however, that simulations may experience decreased speed and increased memory requirements when dealing with objects with a high number of vertices.
>
> *We wish that our response has addressed your concerns, and turns your assessment to the positive side. If you have any more questions, please feel free to let us know during the rebuttal window.*
>
> Best,
> Authors

---

> > ### Comment · Reviewer_Vd5A · 2023-11-21
> >
> > Thank you for the detailed comments and additional information provided, it definitely improved my view of the paper. While it might be hard to fit all the additional detail into the main text of the paper I would to see some of these at least alluded to with more detail being provided in the appendix.

---

> > > ### Author Response · Authors · 2023-11-21
> > >
> > > We are pleased to hear that it has contributed positively to your perception of the paper. The additional details have been incorporated into the appendix in the revised version. With the rebuttal period concluding shortly, we hope our response adequately addresses your questions and concerns. If so, would you kindly consider adjusting the score accordingly? Again, thank you for your thoughtful review and constructive feedback.

---

> ### Author Response · Authors · 2023-11-21
> **Supplementary Simulation Details**
>
> We provide additional details about the implementation of our simulation, specifically addressing the process and methods we used to compute the derivatives and Hessian matrix to address your concerns.
>
> **1. Contact Energy**
> According to Section 3.2, the contact energies $U_r$ and $U_f$ are defined as
> $$
> U_r(d)=\frac{k_r}2\max(d-\epsilon_r, 0)^2,U_f(\mathbf u)=\mu\lambda f_0(\epsilon_v,\|\mathbf u\|),
> $$
>
> where the distance $d$ is a signed distance between vertex $\mathbf p$ and its projection to a triangle $(\mathbf p_1,\mathbf p_2,\mathbf p_3)$, whose normal direction is represented by $(\mathbf p_2-\mathbf p_1)\times(\mathbf p_3-\mathbf p_1)$. $k_r$ stands for a constant stiffness parameter, $\mu$ stands for a friction coefficient, and $\lambda$ stands for the contact force along normal direction at last timestep. The signed distance $d$ is then calculated as
> $$
> \begin{aligned}
> d=\frac{(\mathbf p-\mathbf p_1)\cdot[(\mathbf p_2-\mathbf p_1)\times(\mathbf p_3-\mathbf p_1)]}{\|(\mathbf p_2-\mathbf p_1)\times(\mathbf p_3-\mathbf p_1)\|}\\
> =\frac{(\mathbf p-\mathbf p_1)\cdot\det
> [\mathbf p_2-\mathbf p_1,\mathbf p_3-\mathbf p_1,(\hat{\mathbf x},\hat{\mathbf y},\hat{\mathbf z})]}{\|(\mathbf p_2-\mathbf p_1)\times(\mathbf p_3-\mathbf p_1)\|}\\
> =\frac{\det[\mathbf p_2-\mathbf p_1,\mathbf p_3-\mathbf p_1,\mathbf p-\mathbf p_1]
> }{\|(\mathbf p_2-\mathbf p_1)\times(\mathbf p_3-\mathbf p_1)\|},
> \end{aligned}
> $$
>
> in which the 3D vertices are represented as column vectors. Let $\mathbf x=[\mathbf p_1,\mathbf p_2,\mathbf p_3,\mathbf p]\in\mathbb R^{12}$ denote the involved variables, we implemented tool functions in Taichi to calculate
> $$
> \begin{aligned}
> D(\mathbf x)&=\det[\mathbf p_2-\mathbf p_1,\mathbf p_3-\mathbf p_1,\mathbf p-\mathbf p_1]
> ,\\
> C(\mathbf x)=\|(\mathbf p_2-\mathbf p_1)\times(\mathbf p_3-\mathbf p_1)\|,
> \end{aligned}
> $$
>
> together with the gradients $\partial D/\partial\mathbf x,\partial C/\partial\mathbf x$ and Heissan matrices $\partial^2 D/\partial \mathbf x^2,\partial^2 C/\partial\mathbf x^2$. With the help of SymPy library for symbolic computation, we optimized the computation graph and hard coded it into Taichi snippets. The repulsive energy $U_r$ is then a composition function on $D(\mathbf x),C(\mathbf x)$, and therefore we use the chain rule to calculate its gradient and Hessian matrix.
>
> For the friction energy $U_f(\mathbf u)$, we follow IPC [1] to calculate the $f_0(\epsilon_v,\|\mathbf u\|)$ function, its derivative $f_1(\epsilon_v,\|\mathbf u\|)$ and second derivative $f_2(\epsilon_v,\|\mathbf u\|)$. Once we get the $\partial U_f/\partial \|\mathbf u\|$ and $\partial^2 U_f/\partial \|\mathbf u\|^2$, we transform it back to the tangential space to get $\partial U_f/\partial\mathbf u$ and $\partial^2U_f/\partial\mathbf u^2$.
>
> **2. Elastic Energy**
>
> We follow Stable Neo-Hookean [2] to define the elastic energy as
> $$
> U_e(\mathbf F)=\frac{\mu}{2}(I_C-3)-\mu(J-1)+\frac\lambda2(J-1)^2,
> $$
> in which $\mathbf F$ denote the deformation gradient of one tetrahedral element, $I_C=\text{tr}(\mathbf F^T\mathbf F)$ and $J=\det \mathbf F$. The motivation of replacing the original $\log J$ term in original Neo-Hookean model to $J-1$ is that $J-1$ is stable even for $J\le0$. Therefore, even if the whole tetrahedral element is inverted, this Stable Neo-Hookean model still works robustly and can provide gradients & Hessian matrices to recover the element (See the original paper of Stable Neo-Hookean [2]). This situation is common in our simulator, since the robot manipulator involves a moving boundary and can easily cause invertion when the moving speed is large. We observe that Stable Neo-Hookean perfectly addresses this problem.
>
> The gradient $\partial U_e/\partial \mathbf F$ can be computed easily. To compute the Hessian matrix $\partial ^2U_e/\partial\mathbf F^2$, we first enumerate the differential $\delta F_{ij}$ for single entry and then calculate the correponding force differential $\delta(\partial U_e/\partial\mathbf F)$. We follow a well-known tutorial on FEM [3] to calculate the force differentials, and then concatenate them into the Hessian matrix $\partial^2 U_e/\partial \mathbf F^2$.

---

> ### Author Response · Authors · 2023-11-21
> **Supplementary Simulation Details Part 2**
>
> **3. Thin-Shell Energy**
>
> Following [4], we apply the stretching energy $U_s$ and the bending energy $U_b$ for thin-shell materials. We further divide the stretching energy into $U_s^e$ and $U_s^a$, representing stretching energy for edges and areas of triangles. The energy formula is as follows:
> $$
> \begin{aligned}
> U_s^e=\sum_{e}K_e(1-|e|/|\overline{e}|)^2|\overline{e}|,
> U_s^a=\sum_{A}K_a(1-|A|/|\overline{A}|)^2|\overline{A}|,
> U_b=\sum_{e}K_b(\theta_e-\overline{\theta_e})^2|\overline{e}|/|\overline{h_e}|,
> \end{aligned}
> $$
> where $e$ represents all the edges, $A$ represents all the triangles, and $\theta_e$ represents the angle on edge $e$. $|\overline{e}|$, $|\overline{A}|$, $|\overline{h_e}|$, and $\overline{\theta_e}$ respectively represent the rest length of the edge, the rest area for the triangle, one-third of the average height of the two adjacent triangles of the edge, and the rest angle on this edge. $K_e$, $K_a$, and $K_b$ are the coefficients.
>
> For $U_s^e$, it's simple to calculate the derivatives and Hessian matrix. The derivative of the length $|e|$ to the nodal point positions $x_0$ and $x_1$ is $\frac{\partial |e|}{\partial x_0}=(x_0 - x_1)/|e|$. The Hessian matrix will require the calculation of $\frac{\partial^2 |e|}{\partial a_0^2}$ and $\frac{\partial^2 |e|}{\partial a_0 \partial b_0}$ due to symmetry, where $x_0=(a_0,b_0,c_0)$.
>
> For $U_s^a$, it becomes more complex to compute the derivatives and Hessian matrix. Assume the triangle is $(x_0, x_1, x_2)$ and $x_0=(a_0, b_0, c_0)$. Similar to the calculation of the contact energy, we use math tools to generate the symbolic formula of the derivative $\frac{\partial |A|}{\partial a_0}$ and hard code it with Taichi; then, we can calculate all the derivatives due to symmetry. For the Hessian matrix, we use math tools to generate formulas for $\frac{\partial^2 |A|}{\partial a_0^2}$, $\frac{\partial^2 |A|}{\partial a_0 \partial a_1}$, $\frac{\partial^2 |A|}{\partial a_0 \partial b_0}$, $\frac{\partial^2 |A|}{\partial a_0 \partial b_1}$; then, we can calculate the full matrix using symmetry.
>
> For $U_b$, the Hessian matrix is too complex to compute, even when we try to use some math tools. As a result, we follow [5] to compute the derivatives and Hessian matrix for bending energy. This paper discusses how to compute the Hessian matrix of bending energy in a simplified form.

---

> ### Author Response · Authors · 2023-11-21
> **Supplementary Simulation Details Part 3**
>
> **4. Back-Propagation**
>
> This part has been briefly discussed in Appendix A.2. The equation to be solved in forward simulation is
> $$
> \mathbf 0=\nabla \mathbf g(\mathbf x) = \frac 1{2h^2}\mathbf M(\mathbf x - 2 \mathbf x^{t} + \mathbf x^{t-1} - h^2\mathbf M^{-1}\mathbf f_{ext}) - \mathbf f_{int}(\mathbf x, \mathbf x_{fixed}^t, \mathbf x^t, \overline{\mathbf \theta}^t,\mathbf \eta), (1)
> $$
> where $\mathbf x^t$ and $\mathbf x^{t-1}$ are node positions of $t_{th}$ and $(t-1)_{th}$ timestep,
>
> $\overline{\mathbf \theta}^t$ are undeformed angles for $t_{th}$ timestep,
> $\mathbf x_{fixed}^t$ are positions of fixed nodes in the beginning of $(t+1)_{th}$ timestep,
>
> and $\mathbf\eta$ is the material parameters.
> Here, $\overline{\mathbf \theta}^t$ is actually a function of $\mathbf x^{t}$ and $\overline{\mathbf \theta}^{t - 1}$, and the friction force is a function of $\mathbf x_{i}$ and $\mathbf x_{i - 1}$. DiffPD [6] discusses the general idea of how to pass gradient in the implicit time integration:
>
> $$
> \frac{\partial \nabla \mathbf g(\mathbf x)}{\partial \mathbf x}\frac{\partial \mathbf x}{\partial \mathbf y} + \frac{\partial \nabla \mathbf g(\mathbf x)}{\partial \mathbf y} = \frac{\partial }{\partial \mathbf y}\mathbf 0
> $$
> $$
> \frac{\partial }{\partial \mathbf x}[\frac{1}{h^2}\mathbf M(\mathbf x-\mathbf y)+\nabla \mathbf E(\mathbf x)]\frac{\partial \mathbf x}{\partial \mathbf y} + \frac{\partial }{\partial \mathbf y}[\frac{1}{h^2}\mathbf M(\mathbf x-\mathbf y)] = 0
> $$
> $$
> [\frac{1}{h^2}\mathbf M + \nabla^2 \mathbf E(\mathbf x) ]\frac{\partial \mathbf x}{\partial \mathbf y} - \frac{1}{h^2}\mathbf M = 0
> $$
> $$
> \frac{\partial \mathbf x}{\partial \mathbf y} = \frac{1}{h^2}[\nabla^2 \mathbf g(x)]^{-1}\mathbf M
> $$
> With $\frac{\partial \mathbf x}{\partial \mathbf y}$ in hand, we can backpropagate the gradient from $x^{t+1}$ to $y^t$. Given $y^t=2x^t-x^{t-1}+h^2M^{-1}f_{ext}$, we can deliver the gradient to both $x^t$ and $x^{t-1}$.
>
> Similarly, by substituting $y$ in the above equation with other variables like $\mathbf x_{fixed}^t$ or $\overline{\mathbf \theta}^t$, we can calculate the gradient. The remaining challenge is computing the term $\frac{\partial \nabla \mathbf g(\mathbf x)}{\partial \mathbf y}$ in the first line when changing $y$ into other variables. For these variables, this term will involve the Jacobian matrix for the force, as shown in Equation (1).
>
> For $\overline{\mathbf \theta}^t$ and $\mathbf\eta$, their relationship to the forces is linear, allowing us to calculate the Jacobians directly. Regarding $\mathbf x_{fixed}^t$, the Jacobians are computed during forward simulation, where they are set to zeros. We can apply the same procedure in forward simulation and retain the Jacobians during the backpropagation process.
>
> Concerning $\mathbf x^t$, it primarily influences the friction force since we utilize the pressure force, contact surface, and the original projected point from the last timestep. We observe that [7] calculates the derivative for friction similarly to our approach. The distinction lies in our choice not to pass the gradient through the surface change during backpropagation
>
> *Again, thank you for your insightful and constructive comments!*
>
> [1] Li, Minchen, et al. "Incremental potential contact: intersection-and inversion-free, large-deformation dynamics." ACM Trans. Graph. 39.4 (2020): 49.
>
> [2] Smith, Breannan, Fernando De Goes, and Theodore Kim. "Stable neo-hookean flesh simulation." ACM Transactions on Graphics (TOG) 37.2 (2018): 1-15.
>
> [3] Eftychios Sifakis and Jernej Barbic. Fem simulation of 3D deformable solids: A practitioner’s guide to theory, discretization and model reduction. ACM SIGGRAPH 2012 Courses, 2012. 4
>
> [4] Grinspun, Eitan, et al. "Discrete shells." Proceedings of the 2003 ACM SIGGRAPH/Eurographics symposium on Computer animation. 2003.
>
> [5] Tamstorf, Rasmus, and Eitan Grinspun. "Discrete bending forces and their Jacobians." Graphical models 75.6 (2013): 362-370.
>
> [6] Du, Tao, et al. "Diffpd: Differentiable projective dynamics." ACM Transactions on Graphics (TOG) 41.2 (2021): 1-21.
>
> [7] Huang, Zizhou, et al. "Differentiable solver for time-dependent deformation problems with contact." arXiv preprint arXiv:2205.13643 (2022).

---

### Author Response · Authors · 2023-11-18
**General Response**

# General Response to All Reviewers
We express our gratitude to all the reviewers for their perceptive comments and helpful suggestions aimed at enhancing the quality of our work. Beyond addressing individual reviewers' feedback, we would also like to emphasize our contributions and introduce new experiments in our rebuttal.

**1. Our Contributions**
We are glad to find out that the reviewers generally acknowledge our contributions:
* We have constructed a comprehensive benchmark focusing on the manipulation of thin-shell materials, reflecting the inherent properties of such materials and emphasizing specific challenges.
* We implement a fully differentiable simulation engine, specifically customized for robotic manipulation involving thin-shell materials.
* We involve benchmarking the performance of various methods and introduces a hybrid approach to effectively address the challenges associated with thin-shell material manipulation.

**2. New Experiments**
In this rebuttal, we have added more supporting experiments to address reviewers’ concerns.

* **[A]** We execute a set of human-designed policies to further gauge the success levels indicated by the reward values in each task.
* **[B]** We introduce a novel real-world experiment involving paper folding to further showcase our sim-to-real ability.

**3. Simulation Details**
We observe that most of questions arise from the implementation details of our physics simulation engine [Vd5A, 3BVm, k5Lf]. Here we give an detailed description on how the simulator functions. In each timestep $t$ of the forward simulation, the simulator sequentially perform the following calculations:

1. Apply the action of current frame, which is obtained from a neural network or directly from an action sequence. (Section 4.1)
2. The potential collision pairs are detected and stored as a constraint set. (Section 3.2)
3. Reset the Hessian matrix $\partial^2U/\partial\mathbf x^2$, force $-\partial U/\partial \mathbf x$ and energy $U(\mathbf x)$ of system to zero.
4. Calculating and summing up all energies $U_k(\mathbf x)$, forces $-\partial U_k/\partial \mathbf x$ and Hessian matrices $\partial^2U_k/\partial\mathbf x^2$ of individual nodes and elements in parallel. To guarantee the convergence, we project all Hessian matrices to a manifold of SPD matrices using a parallelized QR iteration.
5. The augment direction
$$
\Delta\mathbf x=-(\partial^2g/\partial\mathbf x_t^2)^{-1}(\partial g/\partial \mathbf x_t),\text{where }g(\mathbf x)=\frac{1}{2\Delta t^2}\|\mathbf x-\mathbf y_t\|_{\mathbf M}+U(\mathbf x)
$$
is calculated through a parallelized linear equation solver provided by cuSPARSE. (Appendix A.1)
6. Perform a line search to find out the minimal $k\ge0$ that
$$g(\mathbf x_t+2^{-k}\Delta \mathbf x)<g(\mathbf x_t).$$
7. The coordinates is updated by $\mathbf x_{t+1}:=\mathbf x_t+2^{-k}\Delta\mathbf x$. If $\|\Delta\mathbf x\|$ is small enough or the maximum number of iterations has been reached, stop the Newton's iteration. Otherwise, go back to step 2.
8. Update the system velocities by $\mathbf v_{t+1}=(\mathbf x_{t+1}-\mathbf x_t)/\Delta t$.

As illustrated in Section 3.1 and 3.2, the energy $U$ in step 3 can be expressed as $U = U_e + U_s + U_b + U_r + U_f$, where

- $U_e$ is the energy of elastic objects. (Section 3.1)
- $U_s$ is the stretching energy of thin-shell materials. (Section 3.1)
- $U_b$ is the bending energy of thin-shell materials. (Section 3.1)
- $U_r$ is the repulsive penalty energy. (Section 3.2)
- $U_f$ is the friction potential. (Section 3.2)

The 1st and 2nd order derivatives of all energy terms are symbolically calculated and implemented in Taichi.

---

> ### Author Response · Authors · 2023-11-18
> **General Response part 2**
>
> **4. Parallelization and Running Speed of Simulation**
>
> In ThinShellLab, both the forward and backward passes are parallelized in a single GPU. The major bottlenecks of the running time are **1) linear equation solving at each iteration of Newton's method** and **2) Hessian matrices building and SPD projecting**. For Hessian matrices, we use Taichi programming language to process the submatrices for independent elements in parallel. We also implement the QR iteration in Taichi to project each submatrix onto a manifold of SPD (symmetric positive definite) matrices. For linear equation solving, we use a python binding of the cuSPARSE library, which supports solving sparse matrices using Cholesky Factorization on GPU. The rest parts of simulation (e.g., line search and collision detection) are fast enough and therefore can be negligible. The collision detection is efficiently solved by space hashing using Taichi's sparse data structure.
>
> To benchmark the running speed of simulation, we test our simulator on the *Forming* task, which contains a typical scene with 1698 DoFs involving both volumetric and thin-shell deformable bodies. The test is done on a single RTX 4090. In average, the forward / backward passes take **36.41 s / 4.67 s** respectively for 100 timesteps, i.e., framerates of **2.74 fps / 21.41 fps**. In comparison, DiffCloth reports their framerates at 0.73 fps / 2.72 fps using the same Newton-Cholesky method and 7.14 fps / 17.54 fps using the fast PD acceleration, both under a scene with 2469 DoFs. C-IPC reports a framerate at 0.024 fps for forward simulation of a thin-shell scene with 18.3K DoFs.
>
> In practice, it takes about 3 hours to train a policy in one task.
>
> While we run our experiments on a single GPU, it is also possible to run multiple environments on different GPUs in parallel. We didn't do this because gradient-based trajectory optimization method do not benefit from parallel environments.
>
> **5. Details and analysis of hybrid method**
> We list the detailed design and analysis of the hybrid method as follows.
> * **Parameters** We opt for a population size of $40$ in CMA-ES with zero-initialized trajectories. The initial variance is set to 1.0, corresponding to 0.0003 in each dimension of manipulator movement per timestep. The maximum movement within one timestep is either 0.001 or 0.002, depending on the task. Generally, we execute CMA-ES for 80% of the total episodes, where episodes are defined by the number of rollouts. For the Pick-Folding task, which demands more extensive training, we set the CMA-ES episode count to 1000 and supplement it with an additional 150 episodes of gradient descent.
> * **Design** To enhance the effectiveness of the CMA-ES method, we filter out non-feasible states, such as instances where there is no contact between manipulators and thin shell or when the manipulator is experiencing excessive force. In these situations, we directly assign a low reward value and save the time for executing the rest actions.
> * **Analysis** Some curves in Figure 3 exhibit large variance. Specifically, in **Folding-U**, **Following**, and **Separating**, the variance arises from the sensitivity of thin-shell manipulation results to minor changes, as previously discussed in the introduction section. In the case of **Pick-Folding**, although sensitivity contributes to the variance, the primary factor is occasional failure of the CMA-ES method. Failures result in rewards near 0, and successful runs yield rewards greater than 6, leading to substantial variance in the outcomes.

---

> ### Author Response · Authors · 2023-11-18
> **General Response part 3**
>
> **6.Success indicator**
>
> Defining the level of success in certain tasks proves challenging due to uncertainty regarding the optimal performance achievable within the current settings using the best policy. Currently, we gauge the success of a policy by visually inspecting the results. Nevertheless, we aim to enhance analytical insight by presenting the scores of human-designed heuristic policies alongside theoretically optimal scores. Subsequently, we can formulate a success indicator based on an analysis of these scores.
>
> * **[Human Designed]** To elucidate the meaning of reward values, we present the scores of human-designed policies for each task in the table below. Notably, generated policies occasionally outperform human-designed counterparts, particularly in tasks requiring meticulous manipulation and where certain policies exhibit unexpected effectiveness (as observed in **Separating**).
> * **[Oracle]** In addition to human-designed policies, we also present the theoretically maximum scores for each task, disregarding certain physical constraints. For **Pick-Folding**, we calculate this score by assuming the entire middle curve is fully folded. In the case of **Separating**, we assume the paper is affixed to the manipulator, while the block experiences no friction force with the paper. For **Following**, we assume both the manipulator and the block are affixed to the paper. In both **Folding-L** and **Folding-U**, we assume that the target curve is fully folded while the other one is completely flattened. For tasks involving **Lifting** and **Forming**, we assume a perfect match with the target. It is important to note that although certain Oracle scores, such as those for **Pick-Folding** and **Separating**, are exceptionally high, they are **not physically feasible within the current experimental settings**. We present this information solely to illustrate the approximate score range for our tasks.
> * **[Success Score]** To make it further concrete, we design the score of success for each tasks. Concretely, we define the score of success as $min(HumanDesign, Oracle/2)$ for **Pick-Folding, Separating Folllowing** and $Oracle/2$ for **Folding-U** and **Folding-L** since human designed policies perform poorly in those two tasks. And the $-3e-4$ as the successful score of **Forming** and **Lifting**.
> * **[Success Rate]** Finally, we calculate the success rate of our CMA-ES + GD method according to the mean and standard deviation.
>
> Table A.
> |          | Pick-Folding | Separating | Following | Folding-L | Folding-U | Forming | Lifting |
> | -------- | ---- | ---- | ---- | ---- | ---- | ---- | ---- |
> | Human Design | 5.49 | 7.23 | 5.05 | 1.63 | -0.24 | 0.0 | -0.001 |
> | Oracle | 42.24 | 20.47 | 7.35 | 6.42 | 6.42 | 0.0 | 0.0 |
> | Success | 5.49 | 7.23 | 3.68 | 3.21 | 3.21 | -0.0003 | -0.0003 |
> | Success rate | 0.36 | 1.0 | 0.39 | 1.0 | 1.0 | 0.76 | 1.0 |
>
> **7. Conclusion and future directions**
> Through the evaluation of various methods in our benchmark tasks of thin-shell manipulation, we have concluded that reinforcement learning algorithms exhibit limited data efficiency. This inefficiency arises from the sensitivity of thin-shell materials and the heightened complexity of their states. Simultaneously, gradient-based methods face challenges in reliability within contact-rich scenarios.While sample-based methods, such as CMA-ES, generally perform well, they suffer from reduced data efficiency in later episodes and intricate settings. Consequently, we propose a hybrid approach that combines the strengths of CMA-ES and gradient-based methods to achieve superior results.
>
> In terms of future work, we recommend exploring closed-loop control under varied parameters, initializations, and tasks. Currently, our focus has primarily been on open-loop methods like CMA-ES and gradient-based methods. We posit that an avenue worth exploring involves leveraging the synergy between reinforcement learning and gradient methods [1] for improved data efficiency in closed-loop control. Additionally, an alternative strategy could involve collecting successful trajectories under diverse settings using open-loop methods and subsequently employing a neural network for imitation learning. This approach aims to address generalization challenges rather than optimization problems.
>
> We hope our responses below convincingly address all reviewers’ concerns. We thank all reviewers’ time and efforts again!
>
> [1] Mora, Miguel Angel Zamora, et al. "Pods: Policy optimization via differentiable simulation." International Conference on Machine Learning. PMLR, 2021.

---

### Meta-Review · Area_Chair_a91Z · 2023-12-03

**Metareview:**

The paper unanimously received Accept ratings from the reviewers. Initially, there were concerns among the reviewers regarding the clarity of the simulator's description, the visualizations, and the placement of crucial details in the Appendix. However, the rebuttal effectively addressed these concerns. The AC finds the simulator particularly interesting as it delves into a compelling problem across diverse task settings. Therefore, the AC concurs with the reviewers' recommendation and supports the acceptance of the paper.

**Justification For Why Not Higher Score:**

There are two main issues: (1) The simulator operates at a relatively slow speed (2-3 Hz), which is slower than real counterparts. This contradicts one of the primary purposes of having a simulator. Typically, we use simulators because they are much faster than reality, enabling large-scale training. (2) While the tasks are diverse, the object shapes are not.

**Justification For Why Not Lower Score:**

The paper tackles an important concept in object manipulation, offering valuable insights that cater to a broad audience.

---

### Decision · Program_Chairs · 2024-01-16

Accept (spotlight)